# Molecular mechanisms that stabilize short term synaptic plasticity during presynaptic homeostatic plasticity

**Jennifer M Ortega, Özgür Genç, Graeme W Davis\***

Department of Biochemistry and Biophysics, Kavli Institute for Fundamental Neuroscience, University of California San Francisco, San Francisco, California

**Abstract** Presynaptic homeostatic plasticity (PHP) compensates for impaired postsynaptic neurotransmitter receptor function through a rapid, persistent adjustment of neurotransmitter release, an effect that can exceed 200%. An unexplained property of PHP is the preservation of short-term plasticity (STP), thereby stabilizing activity-dependent synaptic information transfer. We demonstrate that the dramatic potentiation of presynaptic release during PHP is achieved while simultaneously maintaining a constant ratio of primed to super-primed synaptic vesicles, thereby preserving STP. Mechanistically, genetic, biochemical and electrophysiological evidence argue that a constant ratio of primed to super-primed synaptic vesicles is achieved by the concerted action of three proteins: Unc18, Syntaxin1A and RIM. Our data support a model based on the regulated availability of Unc18 at the presynaptic active zone, a process that is restrained by Syntaxin1A and facilitated by RIM. As such, regulated vesicle priming/super-priming enables PHP to stabilize both synaptic gain and the activity-dependent transfer of information at a synapse.
DOI: https://doi.org/10.7554/eLife.40385.001

## Introduction

Presynaptic homeostatic plasticity (PHP) is an evolutionarily conserved form of homeostatic control that is expressed in organisms ranging from fly to human (*Cull-Candy et al., 1980*; *Plomp et al., 1992*; *Davis, 2013*; *Wang et al., 2011*), at both central and peripheral synapses (*Liu and Tsien, 1995*; *Davis and Goodman, 1998*; *Burrone et al., 2002*; *Thiagarajan et al., 2005*; *Kim and Ryan, 2010*; *Zhao et al., 2011*; *Davis, 2013*; *Henry et al., 2012*; *Jakawich et al., 2010*). PHP can be induced in less than ten minutes and is expressed as a dramatic increase in synaptic vesicle fusion ($\geq$200%) at a fixed number of presynaptic release sites.

An unexplained, emergent property of PHP is the preservation of short-term release dynamics during a stimulus train, referred to here as short-term plasticity or 'STP' (*Figure 1*; see *Weyhersmüller et al., 2011*; *Müller et al., 2012b*; *Müller et al., 2015*; *Orr et al., 2017*). STP is a fundamental property of neural coding, underlying behaviorally relevant circuit-level computations (*Davis and Murphey, 1994*). Indeed, STP is described as being '...an almost necessary condition for the existence of (short-lived) activity states in the central nervous system' (*Von der Malsburg, 1986*; as recently quoted in *Taschenberger et al., 2016*). Thus, the fact that STP is held constant during the expression of PHP may be essential to the life-long stabilization of neural circuit function and animal behavior. But, it remains fundamentally unknown how presynaptic release can be rapidly doubled at a fixed number of active zones while maintaining constant short-term release dynamics.

Two processes within the presynaptic terminal are known to be required for the homeostatic potentiation of vesicle release: 1) an increase in presynaptic calcium influx controlled by ENaC channel insertion in the presynaptic membrane (*Younger et al., 2013*; *Orr et al., 2017*) and 2) an increase in the readily releasable pool of synaptic vesicles that requires the presynaptic scaffolding

**\*For correspondence:**
graeme.davis@ucsf.edu

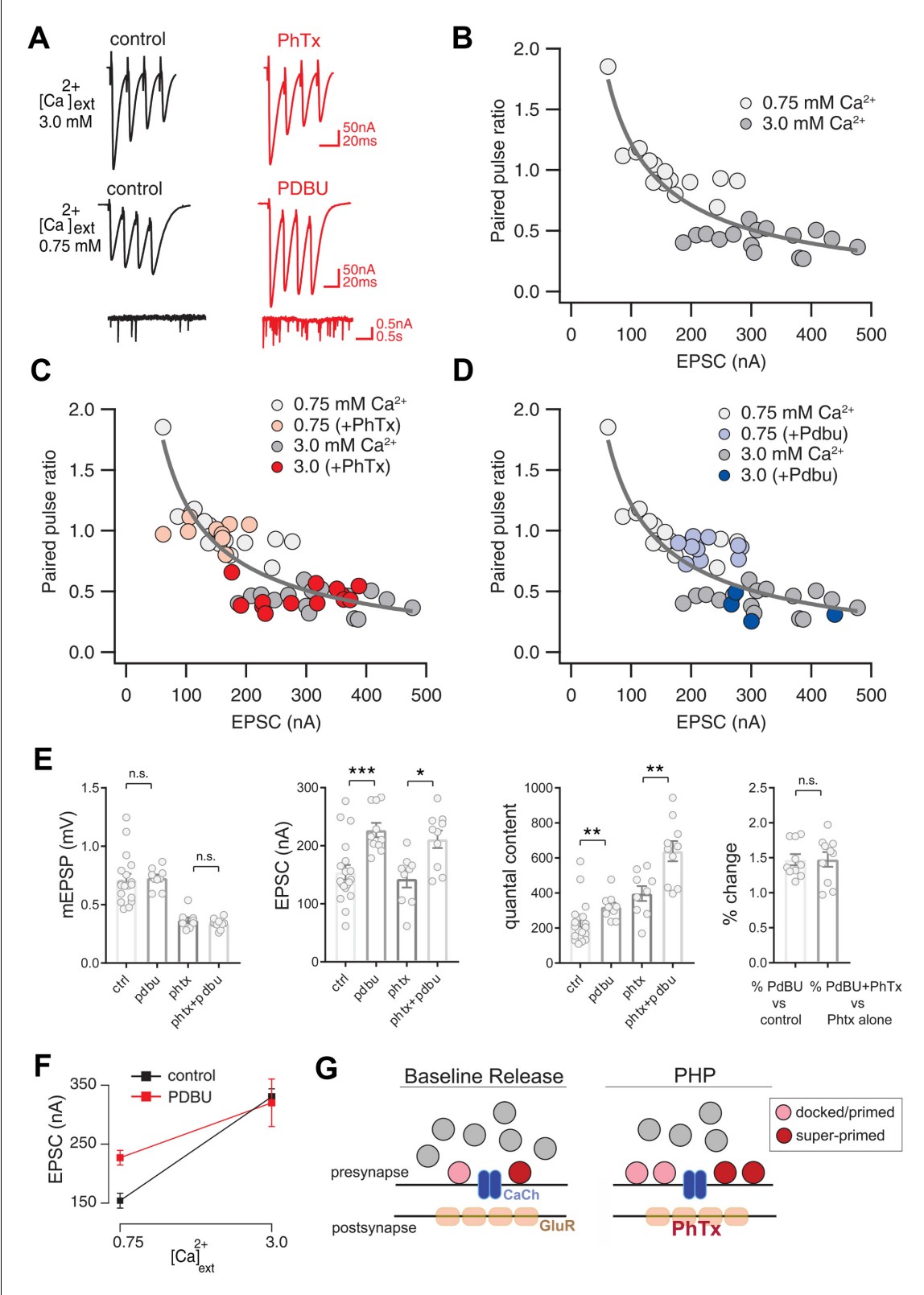

**Figure 1.** Preservation of Release Dynamics During Presynaptic Homeostatic Plasticity. (A) Example traces at the indicated external calcium concentration in the presence or absence of PhTx and PdBu. (B) Paired-pulse ratio (EPSC4/EPSC1) versus initial EPSC amplitude (EPSC1) at two external calcium concentrations as indicated. (C) Data from (B) re-plotted (gray) with the addition of data recorded in the presence of PhTx (light and dark red) for indicated external calcium concentrations. (D) Data from (B) re-plotted (gray) with the addition of data recorded in the presence of PdBu (light and

*Figure 1 continued*

dark blue) for indicated external calcium concentrations. (**E**) Data for mEPSP, EPSC and quantal content for control (ctrl), control in the presence of PdBu (PdBu), control in the presence of PhTx (PhTx) and control synapses incubated in PhTx followed by PdBU (PhTx +PdBu). At right, percent change is calculated as quantal content recorded in the presence of PdBu versus control in the absence of PdBu. (**F**) EPSC amplitude in the presence (red) and absence (black) of PdBu at the indicated extracellular calcium concentrations. (**G**) Schematic highlighting the homeostatic doubling of the pool of docked primed vesicles during presynaptic homeostatic plasticity (PHP) in the presence of PhTx. The ratio of primed (light red) to super-primed (dark red) vesicles is held constant, thereby preserving presynaptic release dynamics. ns, not significant; *p<0.05, **p<0.01, ***p<0.001; Data represent mean ±SEM. Student's t-test, two tailed.

DOI: https://doi.org/10.7554/eLife.40385.002

The following figure supplement is available for figure 1:

**Figure supplement 1.** PdBu-dependent potentiation converges to wild type steady state during a prolonged stimulus train.

DOI: https://doi.org/10.7554/eLife.40385.003

proteins RIM and RBP (*Müller et al., 2012b*; *Davis and Müller, 2015*; *Müller et al., 2015*). It has been proposed that the combined effects of elevated presynaptic calcium and an increased supply of release ready vesicles are sufficient to achieve PHP. But, current models have yet to address how short-term release dynamics are stabilized.

STP can be strongly influenced by the partition of the readily releasable vesicle pool into two functional subclasses: (1) docked vesicles that have a low intrinsic calcium sensitivity of release that are referred to as 'primed' and (2) docked vesicles that have a relatively higher intrinsic calcium sensitivity and are referred to as 'super-primed'. During a stimulus train, super-primed vesicles will dominate release during the first few action potentials and primed vesicle will dominate subsequent release (*Taschenberger et al., 2016*; *Lee et al., 2013*; *Müller et al., 2010*). Thus, a synapse harboring a large proportion of super-primed vesicles will favor a high initial release rate followed by synaptic depression while a synapse harboring a small proportion of super-primed vesicles will be prone to facilitation of release, followed by subsequent synaptic depression (*Taschenberger et al., 2016*; *Lee et al., 2013*). There is both pharmacological and genetic evidence in support of this model. Genetic mutations that impair vesicle super-priming convert depression-prone, high-release probability synapses into low-release probability synapses that express short-term facilitation (*Frank et al., 2006*; *Deák et al., 2009*; *He et al., 2017*). The calcium sensitivity of vesicle fusion is also highly sensitive to phorbol esters (PdBu), which lower the fusion barrier to release (*Taschenberger et al., 2016*; *Lee et al., 2013*). Application of PdBu, which is considered to drive the super-priming process, dramatically potentiates vesicle release and leads to enhanced short-term synaptic depression (*Wang et al., 2016*; *Lee et al., 2013*).

Here, we provide evidence that the stabilization of STP during PHP is achieved by maintaining a constant ratio of primed to super-primed synaptic vesicles. First, we confirm that STP is precisely preserved during the expression of PHP. Second, we document a PdBu sensitive release mechanism at the Drosophila NMJ and provide evidence that the fraction of super-primed vesicles is maintained during the full extent of PHP. Third, we provide molecular insight into how a constant fraction of super-primed vesicles is maintained during the expression of PHP. We demonstrate that Unc18 has an evolutionarily conserved function during the rapid induction of PHP. We show that Unc18 function during PHP is facilitated by the activity of presynaptic RIM and, remarkably, is antagonized by presynaptic Syntaxin. Based upon these and other data, we present a new model for the homeostatic control of synaptic vesicle release that is based upon the regulated control of Unc18 levels at the presynaptic release site, acting in concert with the priming activity of RIM to stabilize STP in the presence of a homeostatic doubling presynaptic neurotransmitter release.

## Results

We begin by documenting how the short-term dynamics of presynaptic release (short-term plasticity or STP) are held constant during the expression of PHP. We rapidly induce PHP by application of sub-blocking concentrations of the glutamate receptor antagonist philanthotoxin (PhTx, 10–20 µM). PhTx causes a ~ 50% decrease in miniature excitatory postsynaptic potential (mEPSP) amplitude and induces a homeostatic increase in presynaptic vesicle release that precisely counteracts the change in mEPSP amplitude, maintaining the amplitude of action potential evoked neurotransmitter release

at baseline levels (*Frank et al., 2006*; *Davis, 2013*). As shown in *Figure 1A*, even at elevated extra-cellular calcium (3.0 mM $[Ca^{2+}]_e$), excitatory postsynaptic currents (EPSC) are precisely maintained at control levels following application of PhTx (see *Müller and Davis, 2012a*; see below for further quantification).

Next, we characterize the expression of STP at two concentrations of external calcium (0.75 mM and 3.0 mM $[Ca^{2+}]_e$), doing so in the presence and absence of PhTx to induce PHP (*Figure 1A–C*). Note that the effects of non-linear summation prevents accurate calculation of presynaptic release at calcium concentrations in excess of approximately 0.3 mM. As such, measurement of synaptic currents in two electrode voltage clamp configuration are essential for experiments where extracellular calcium exceeds 0.3 mM. As expected (*Zucker and Regehr, 2002*), STP is strongly dependent on the concentration of external calcium, showing facilitation at 0.75 mM $[Ca^{2+}]_e$ and depression at 3.0 mM $[Ca^{2+}]_e$ (*Figure 1A,B*). Application of PhTx causes an approximate doubling of presynaptic release during PHP that is similar in magnitude to the change in release observed when comparing wild type neurotransmission at 0.75 mM $[Ca^{2+}]_e$ and 3.0 mM $[Ca^{2+}]_e$. However, the expression of PHP occurs without a change in release dynamics (*Figure 1B,C*). Specifically, there is no statistically significant change in paired pulse ratio comparing the presence and absence of PhTx, and this is true at both 0.75 mM $[Ca^{2+}]_e$ where facilitation dominates (p = 0.65; Student's t-test, two tailed) and at 3.0 mM $[Ca^{2+}]_e$ where depression dominates (*Figure 1C*; p = 0.45; Student's t-test, two tailed). Thus, PHP is achieved by doubling synaptic vesicle release without altering the expression of STP, confirming prior observations in this system (*Weyhersmüller et al., 2011*; *Müller et al., 2012b*; *Müller et al., 2015*; *Orr et al., 2017*). We sought to understand how this effect could be achieved.

As outlined in the introduction, STP is strongly influenced by the partition of the readily releasable vesicle pool into two functional subclasses: (1) docked vesicles that have a low intrinsic calcium sensitivity of release that are referred to as 'primed' and (2) docked vesicles that have a relatively higher intrinsic calcium sensitivity and are referred to as 'super-primed'. A synapse with a large fraction of super-primed vesicles will show synaptic depression while a synapse with a small number of super-primed vesicles will facilitate. We hypothesize that the ratio of primed to super-primed vesicles is somehow maintained during the expression of PHP, allowing for a dramatic potentiation of vesicle release without altering the dynamics of release during a stimulus train. To test this hypothesis, we used phorbol esters to probe the ratio of primed to super-primed vesicles at the *Drosophila* NMJ.

It is well established that phorbol esters (PdBu) decrease the energy barrier to synaptic vesicle fusion, effectively converting the docked/primed vesicle pool into a super-primed, high release probability state (*Lee et al., 2013*; *Taschenberger et al., 2016*). Thus, magnitude of PdBu-dependent potentiation of presynaptic release is proportional to the size of the pool of synaptic vesicles that reside in a docked/primed, but not super-primed state. If PHP-dependent potentiation of presynaptic release preserves the ratio of primed to super-primed vesicles, then the effects of PdBu should be the same prior to and following application of PhTx to the synapse.

We first characterize the use of PdBu at the *Drosophila* NMJ. At 0.75 mM $[Ca^{2+}]_e$, PdBu strongly potentiates both evoked and spontaneous vesicle fusion and converts STP from facilitation to depression (*Figure 1A,D,E*). Specifically, PdBu has no effect on mEPSP amplitude (*Figure 1E*), causes a significant increase in EPSC amplitude (*Figure 1E*) and a corresponding increase in quantal content (*Figure 1E*). We then express the effects of PdBu as a percent change compared to baseline in the absence of PdBu, observing a significant ~140% increase in release (*Figure 1E*, right; p<0.05). Three further effects were also quantified. First, application of PdBu causes a significant decrease in the paired-pulse ratio (*Figure 1D*; p<0.05). Second, we show that presynaptic release converges to a statistically similar steady state in the presence and absence of PdBu during a prolonged stimulus train (*Figure 1—figure supplement 1*; p>0.1, Student's t-test, two-tailed; comparison of final three data points of stimulus train). Third, just as observed at the mammalian central synapses, we demonstrate that the effects of PdBu are dependent on the concentration of extracellular calcium (*Lee et al., 2013*; *Taschenberger et al., 2016*). When recording at elevated extracellular calcium (3 mM $[Ca^{2+}]_e$), the effect of PdBu on EPSC amplitude is absent (*Figure 1F*). These data are consistent with the existence of a finite pool of docked vesicles that are uniformly accessed by action-potential induced release at elevated calcium, rendering PdBu without effect. Thus, PdBu functions comparably in Drosophila and at mammalian central synapses. And, by comparing the effects of PdBu on synaptic transmission at 0.75 mM $[Ca^{2+}]_e$ we can gain an estimate of the fraction of vesicles that exist within the primed versus super-primed state. Specifically, the magnitude of PdBu-mediated

potentiation at 0.75 mM $[Ca^{2+}]_e$ is proportional to the number of vesicles that remain in the primed (*not* super-primed) state.

We next examined the effects of PdBu at synapses previously incubated in PhTx. Once again, at 0.75 mM $[Ca^{2+}]_{e,}$ PhTx causes a decrease in mEPSP amplitude, an increase in quantal content and no change in evoked EPSC amplitude (*Figure 1E*). When PdBu is applied after PhTx, there is no further change in mEPSP amplitude, compared to PhTx alone, as expected (*Figure 1E*). However, we find that application of PdBu enhances EPSC amplitudes in the presence of PhTx compared to controls with or without PhTx. The consequence is that quantal content is significantly increased compared to PdBu alone and compared to PhTx alone. Indeed, quantal content is potentiated 3-fold compared to baseline release in wild type (from approximately 200 vesicles per action potential to 600 vesicles per action potential). However, when we calculate the percent change in release caused by PdBu compared to PhTx alone, we find that release is potentiated by ~140%, the same percentage increase caused by PdBu applied to a wild type synapse (*Figure 1E*). Thus, the proportion of PdBu-sensitive vesicles remains constant following the induction of PHP. Since PHP in a wild type animal can be expressed without a change in short-term plasticity, we propose that the ratio of super-primed to primed vesicles remains constant during expression of PHP (*Figure 1G*).

## Mechanisms that maintain the ratio of primed to super-primed vesicles

We sought to define the underlying molecular mechanisms that might be responsible for maintaining a precise ratio of primed to super-primed vesicles. It is well established that two synaptic proteins, Unc13 and Unc18, participate in the maturation of vesicles from a docked to a primed and, potentially, super-primed state (*He et al., 2017*; *Park et al., 2017*; *Deák et al., 2009*). Recent evidence demonstrates that a mutation that deletes Drosophila *Unc13A* has no effect on the rapid induction of PHP (Martin Mueller personal communication). By contrast, there is prior evidence that Unc18 might participate in the mechanisms of PHP at the rodent NMJ (*Sons et al., 2003*). Therefore, we focused our attention on Unc18.

Unc-18 is a member of the Sec1/Munc-18 family of syntaxin binding proteins, conserved from yeast to human. Unc-18 is an essential component of the macromolecular synaptic vesicle fusion apparatus. At the neuronal synapse, deletion of the Unc-18 orthologues in worm, fly and mice largely abolish both spontaneous and action potential evoked synaptic vesicle release (*Weimer et al., 2003*; *Harrison et al., 1994*; *Verhage et al., 2000*). But, examination of heterozygous (*unc-18–1/+*) mutants has provided some interesting insight into the potential function of Unc-18 in homeostatic plasticity.

Synaptic transmission persists at *Drosophila*, mouse and human synapses in a heterozygous (*unc-18–1/+*) mutant background (*Wu et al., 1998*; *Toonen et al., 2006*; *Patzke et al., 2015*). Synaptic efficacy is diminished in these animals, indicating that the levels of Unc-18 are limiting for evoked neurotransmitter release (*Patzke et al., 2015*; *Wu et al., 1998*; *Toonen et al., 2006*). Thus, the heterozygous mutant is a condition amenable to exploring whether Unc-18 is also limiting for presynaptic forms of neural plasticity. At the mouse NMJ, it was previously shown that expression of PHP is suppressed by approximately 25% in the heterozygous *unc-18–1/+* mutant background (*Sons et al., 2003*). One possibility is that Unc-18–1 represents an evolutionarily conserved interface of homeostatic signaling and the synaptic vesicle fusion apparatus (*Sons et al., 2003*). However, because diminished levels of Unc-18–1 limit presynaptic release at the mouse NMJ, it is equally plausible that loss of Unc-18 simply restricts the full expression of PHP through a ceiling effect. We sought to use the Drosophila system to further explore the function of Unc18 during PHP and test whether Unc18 is a critical component that stabilizes STP during the induction and expression of PHP.

## Rop is required in the rapid induction of presynaptic homeostasis

In *Drosophila*, there is a single neuronally expressed *unc-18* gene, termed *Rop* (**R**as **op**posite). Throughout this paper, from this point onward, we refer to the *Drosophila* gene as *Rop* and orthologues in other species as *unc-18*. Prior genetic analyses isolated and characterized numerous mutations in the *Rop* gene, demonstrating that Rop is essential for spontaneous and evoked synaptic vesicle fusion, in agreement with data from other species (*Harrison et al., 1994*; *Wu et al., 1998*; *Toonen and Verhage, 2007*). We have taken advantage of previously characterized mutations in *Rop* to study the role of Unc-18 in presynaptic homeostatic plasticity (PHP).

We assayed PHP in two independent, heterozygous *Rop* loss-of-function mutants: 1) a previously characterized null allele (*Rop^G27*; *Harrison et al., 1994*; see *Figure 2A*) and 2) a small chromosomal deficiency that deletes the entire *Rop* gene locus (*Df^(3L)BSC735*, referred to hereafter as *Df^Rop*; *Cook et al., 2012*; *Figure 2A*). In both *Rop^G27*/+ and *Df^Rop*/+ heterozygous mutants, PHP is significantly suppressed compared to wild type (*Figure 2B–2I*). Specifically, upon application of PhTx, *Rop^G27*/+ mutants show an enhancement of quantal content (*Figure 2H*; p<0.01), but the magnitude of this enhancement is statistically significantly smaller than that observed in wild type (*Figure 2I*; p<0.001). Consistent with impaired homeostatic plasticity, EPSP amplitudes are significantly reduced in the presence of PhTx compared to baseline EPSP amplitudes (*Figure 2C and G*; p<0.05; p<0.0001). Notably, baseline release, recorded in the absence of PhTx, is unaltered in both *Rop^G27*/+ and *Df^Rop*/+ heterozygous mutants at the concentration of external calcium used in this experiment (0.3 mM $[Ca^{2+}]_e$) (*Figure 2C and G*). Thus, two independently derived heterozygous loss-of-function mutants suppress PHP without an effect on baseline neurotransmission, arguing that PHP is highly sensitive to *Rop* gene dosage.

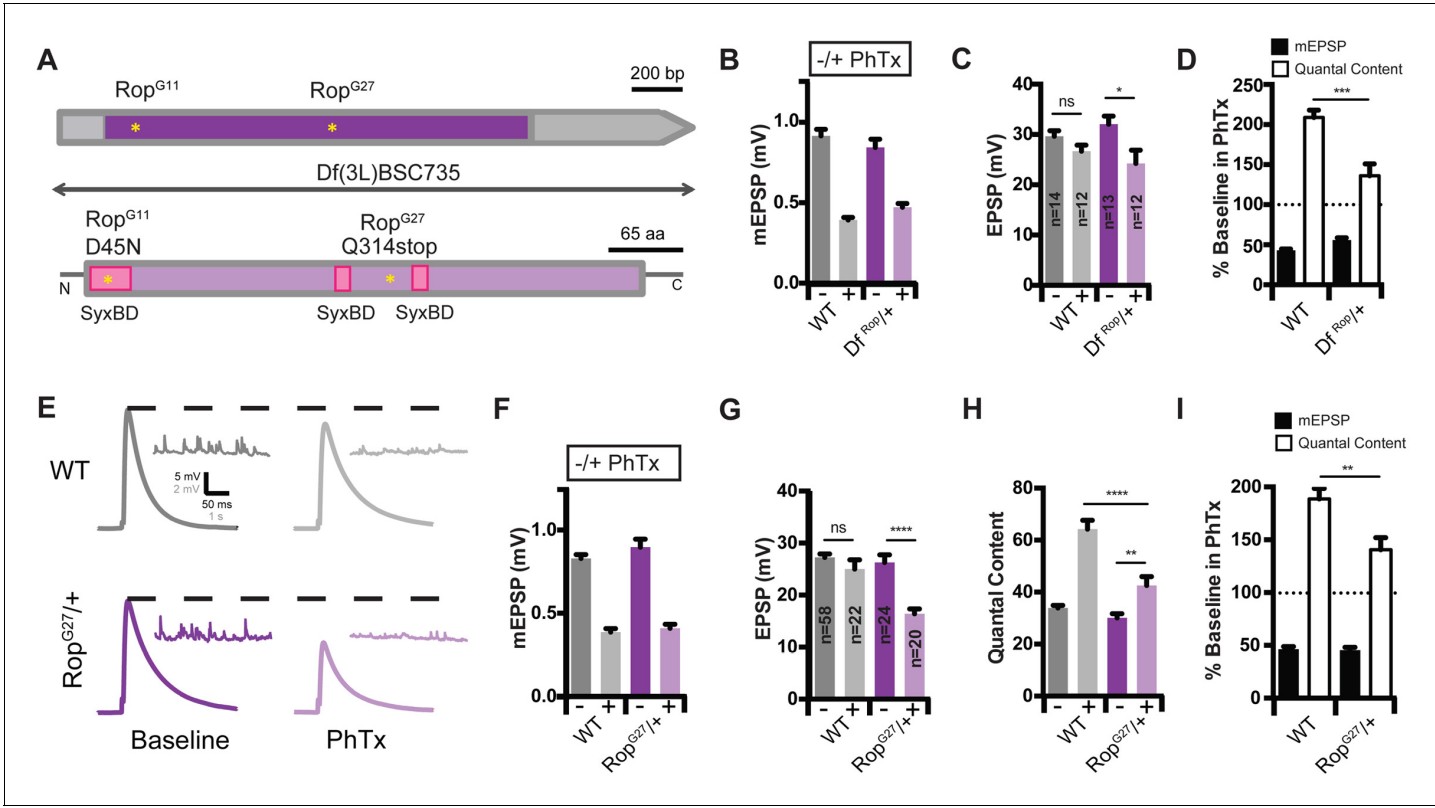

**Figure 2.** Identification of Rop as a SNARE-Associated Molecule Involved in the Rapid Induction of Presynaptic Homeostasis. (A) Schematic of the *Drosophila Rop* gene locus (top) and protein (bottom). Coding exon is shown in dark purple and non-coding DNA is in gray. Protein is shown in light purple. Point mutations in *Rop* mutant alleles (*Rop^G27* and *Rop^G11*) are indicated by yellow stars. Deficiency Df(3L)BSC735 uncovers the *Rop* gene locus as indicated. Syntaxin-binding domains (SBD) of the Rop protein are shown (pink). (B) Average data for mEPSP amplitude in the absence (baseline) and presence (PhTx) of PhTx for WT and heterozygous deficiency chromosome *Df(3L)BSC735* (*Df^Rop*/+). PhTx application reduces amplitudes in all genotypes (p<0.01). Data represent mean ± SEM. (C) Average data for EPSP amplitude as in (B); sample sizes for data in (B–D) are shown on bar graph; ns, not significant; *p<0.05; Student's t-test. (D) Average mEPSP amplitude and quantal content are normalized to values in the absence of PhTx for each genotype. ***p<0.001. (E) Sample traces showing EPSP and mEPSP amplitudes ± PhTx for indicated genotypes. (F–H) Average mEPSP (F) EPSP (G) and Quantal Content (H) for indicated genotypes; ns, not significant; ****p<0.0001; **p<0.01. (I) Average percent change in mEPSP amplitude and quantal content in PhTx compared to baseline for indicated genotypes; **p<0.01. Data are mean ±SEM for all figures. Student's t-test, two tailed.
DOI: https://doi.org/10.7554/eLife.40385.004

## Neuronally expressed rop is required for presynaptic homeostasis

The *unc-18* gene is broadly expressed and has been shown to participate in membrane trafficking events outside the nervous system (*Hata and Südhof, 1995*; *Riento et al., 2000*; *Toonen and Verhage, 2003*). As such, Rop could function either pre- or postsynaptically during PHP. Therefore, we knocked down *Rop* expression specifically in the nervous system using the *Gal4/UAS* expression system. We took advantage of a previously characterized *UAS-Rop-RNAi* transgene (*P{GD1523} v19696*; *Dietzl et al., 2007*) to knock-down *Rop* specifically in neurons (*c155-Gal4*; *Lin and Goodman, 1994*). First, we demonstrate that PHP is strongly suppressed when *UAS-Rop-RNAi* is driven presynaptically (*Figure 3A–E*). Thus, Rop is necessary presynaptically for PHP. We note that presynaptic *Rop* knockdown also causes a decrease in baseline EPSP amplitude and quantal content by ~30% (*Figure 3C*; $p < 0.01$), suggesting that presynaptic knockdown depletes Rop protein levels more substantially than that observed in the heterozygous *Rop* null mutant (*Figure 2F–I*). Indeed, when we compare wild type, $Rop^{G27}/+$ and presynaptic *Rop* knockdown, we find a progressively more severe decrease in mEPSP frequency that is consistent with progressively more severe depletion of Rop protein (*Figure 3F*). Thus, *Rop* knockdown can be considered a strong hypomorphic condition, supporting the conclusion that presynaptic Rop is essential for robust expression of PHP.

## Separable activity of rop during baseline transmission and PHP

Although PHP is selectively impaired in the $Rop^{G27}/+$ heterozygous mutant at 0.3 mM $[Ca^{2+}]_e$, it remains formally possible that loss of Rop places a limit on the number of vesicles that can be released per action potential (quantal content) and, thereby, indirectly restricts the expression of PHP. To address this possibility, we asked whether baseline release and PHP change in parallel as a function of altered extracellular calcium in the $Rop^{G27}/+$ heterozygous mutations. Experiments were conducted at 0.3 mM $[Ca^{2+}]_e$, 0.75 mM $[Ca^{2+}]_e$, 1.5 mM $[Ca^{2+}]_e$ and 3.0 mM $[Ca^{2+}]_e$. First, we find that baseline neurotransmitter release is impaired at 0.75 as well as 1.5 and 3.0 mM $[Ca^{2+}]_e$ in the $Rop^{G27}/+$ heterozygous mutants compared to wild-type (*Figure 4*). But, there remains a highly cooperative relationship between extracellular calcium and vesicle release in the $Rop^{G27}/+$ heterozygous mutants (*Figure 4F*). Next, we demonstrate that PHP is suppressed both at 1.5 mM (*Figure 4A–C*) and 3.0 mM $[Ca^{2+}]_e$ (*Figure 4D–E*), as evidenced by an inability of EPSCs to be restored to baseline values in the presence of PhTx (*Figures 4C* and *3E*) ($p < 0.05$). When we calculate the percent suppression of PHP at 0.3, 1.5, and 3.0 mM $[Ca^{2+}]_e$ we find a constant level of PHP suppression even though baseline release increases ~10 fold over this range of extracellular calcium concentrations (*Figure 4F*). From this we can make two conclusions. First, since release remains sensitive to changes in external calcium, a ceiling effect cannot explain the defect in PHP observed in the $Rop^{G27}/+$ heterozygous mutants. More specifically, the increase in release in $Rop^{G27}/+$ comparing 0.3 mM $[Ca^{2+}]_e$ with 3.0 mM $[Ca^{2+}]_e$ vastly exceeds the change in vesicular release that would be expected for full expression of PHP at 0.3 mM $[Ca^{2+}]_e$ (see *Figure 2*). The fact that PHP is consistently inhibited by ~30%, irrespective of the concentration of external calcium, demonstrates that the magnitude of PHP expression correlates directly with the levels of *Rop* expression, not with the magnitude of evoked release or extracellular calcium concentration, arguing for an essential role of Rop in the mechanisms of PHP.

## Rop dependent vesicle priming correlates with expression of PHP

Unc-18 has been implicated in several different stages of the release process, including vesicle docking, priming, fusion pore formation and regulation of the readily releasable pool (RRP) of synaptic vesicles (*Weimer et al., 2003*; *Toonen et al., 2006*; *Gulyás-Kovács et al., 2007*; *Fisher et al., 2001*; *Toonen and Verhage, 2007*). Since we observe a significant disruption of PHP in the heterozygous $Rop^{G27}/+$ mutant background, we sought to define the parameters of presynaptic release that are particularly sensitive to the heterozygous $Rop^{G27}/+$ mutant, highlighting those actions of Rop that best correlate with impaired PHP. First, we observe a decrease in the frequency of spontaneous mEPSP events in *Rop* mutants (*Figure 3F*). mEPSP frequency is decreased by 36% and 59% compared to wild-type in $Rop^{G27}/+$ and *UAS-Rop-RNAi*, respectively (*Figure 3F*). This is consistent with prior reports in *Drosophila* and other systems (*Wu et al., 1998*; *Toonen et al., 2006*; *Patzke et al., 2015*).

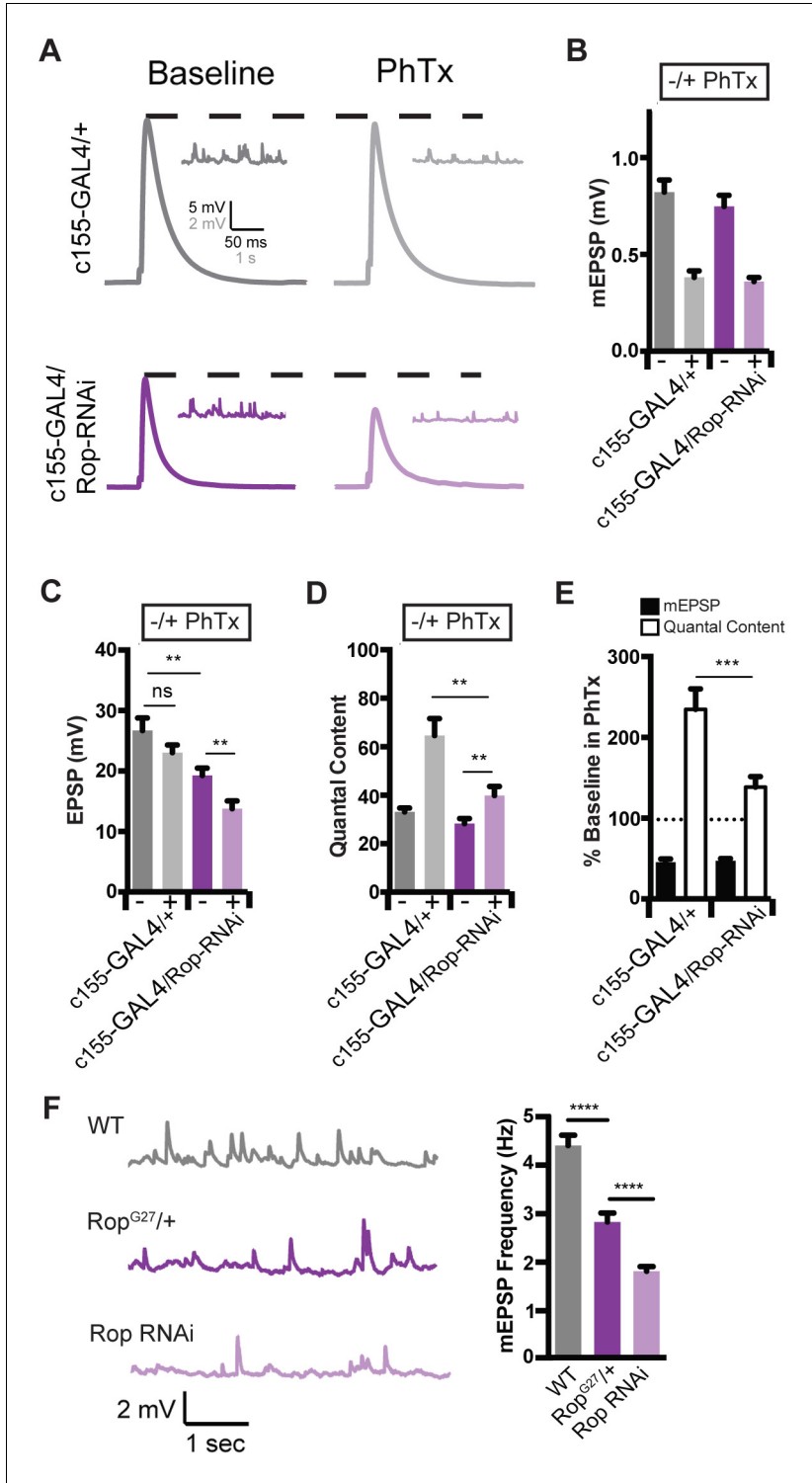

**Figure 3.** Rop is Required Neuronally During Presynaptic Homeostasis. (**A**) Sample traces showing EPSP and mEPSP amplitudes ± PhTx for indicated genotypes. (**B**) Average data for mEPSP when *UAS-Rop-RNAi* is expressed pan-neuronally (*c155-GAL4*)±Phtx as indicated. PhTx reduces amplitudes in all genotypes; p<0.01. (**C**) Average data for EPSP as in B; ns, not significant; **p<0.01. (**D**) Average data for Quantal Content as in B; **p<0.01. (**E**) Average percent change in mEPSP amplitude and quantal content in PhTx compared to baseline for indicated genotypes; ***p<0.001. (**F**) Sample traces showing mEPSPs for indicated genotypes (left) and average mEPSP frequencies (Hz) (right). Student's t-test, two tailed.
DOI: https://doi.org/10.7554/eLife.40385.005

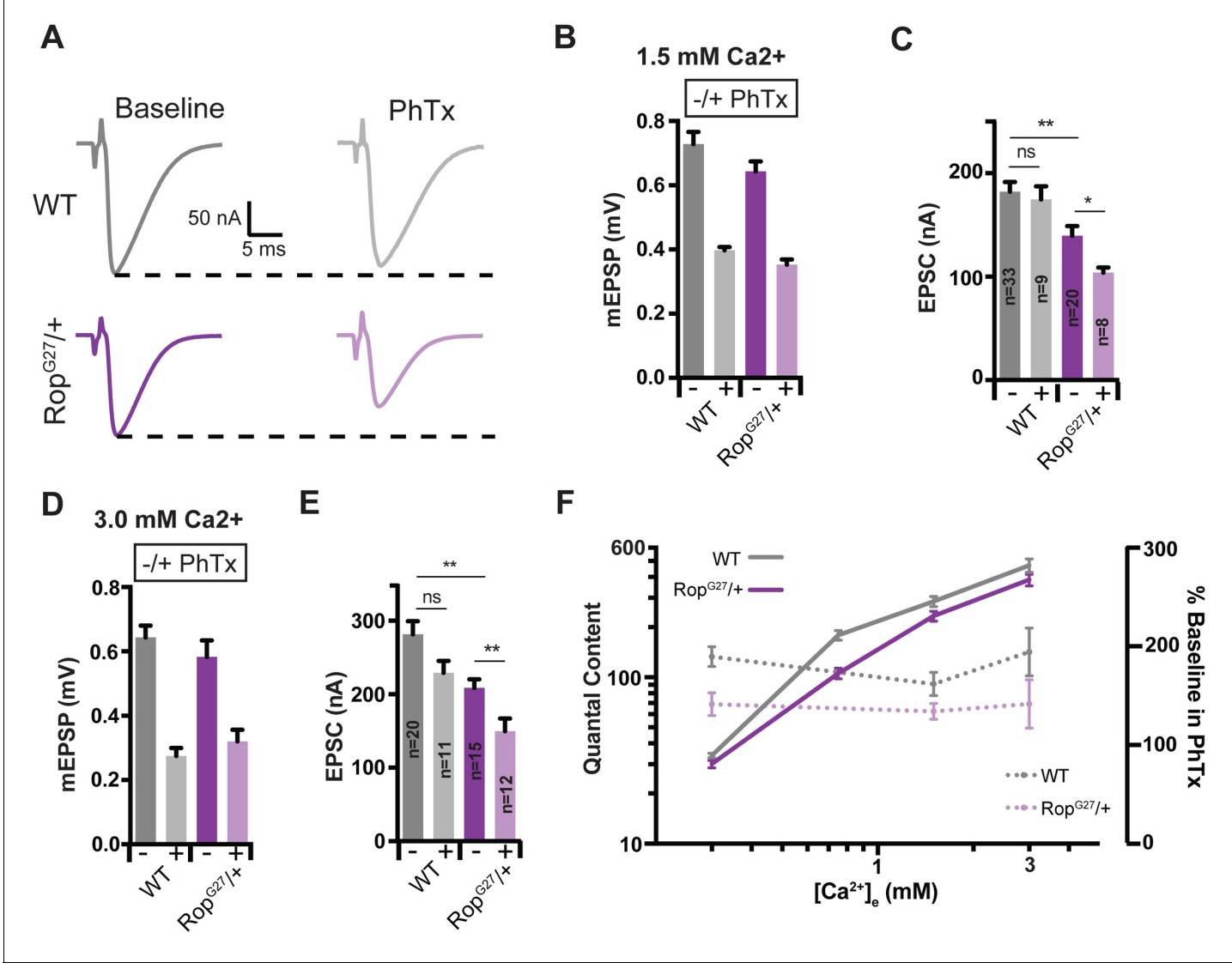

**Figure 4.** Suppression of Presynaptic Homeostasis is maintained with increased $[Ca^{2+}]_e$ in *Rop* mutants. (A) Sample EPSC traces in the absence (baseline) and presence (PhTx) of PhTx for WT and heterozygous $Rop^{G27}$ mutant at 1.5 mM extracellular calcium $[Ca^{2+}]_e$. (B) Average data for mEPSP amplitude ± PhTx for indicated genotypes. PhTx application reduces amplitude in all genotypes (p<0.01). Data represent mean ± SEM. (C) Average data for EPSC amplitude as in (B); ns, not significant; **p<0.01, *p<0.05; Student's t test, two tailed. Sample sizes indicated on bar graph. (D–E) Average mEPSP (D) EPSC (E) ± PhTx for each genotype at 3.0 mM extracellular calcium $[Ca^{2+}]_e$; ns, not significant; **p<0.01; Data represent mean ±SEM. Student's t-test, two tailed. (F) Relationship between mean quantal content and $[Ca^{2+}]_e$ (left axis) and relationship between quantal content normalized to values in the absence of PhTx and $[Ca^{2+}]_e$ (right axis) for WT and $Rop^{G27}$ heterozygous mutants.

DOI: https://doi.org/10.7554/eLife.40385.006

It has previously been shown that a homeostatic modulation of the readily releasable pool (RRP) of vesicles is required for the expression of PHP (*Müller et al., 2012b*). RRP size can be estimated through quantification of the cumulative EPSC amplitude during a high frequency stimulus train (60 HZ, 30 stimuli) at elevated extracellular calcium (1.5 mM; *Figure 5*) and back extrapolation according to published protocols (*Schneggenburger et al., 1999*; *Müller et al., 2015*). We demonstrate that the cumulative EPSC amplitude in the $Rop^{G27}$/+ heterozogous mutant is unaltered at baseline compared to wild-type (*Figure 5B*). Then, we demonstrate that the $Rop^{G27}$/+ mutants show normal modulation of the RRP following application of PhTx, as demonstrated the maintenance of the cumulative EPSC amplitude in the presence and absence of PhTx, which diminishes the amplitude of underlying unitary release events by ~50% (*Figure 5B*). Thus, at the *Drosophila* NMJ, the RRP is not

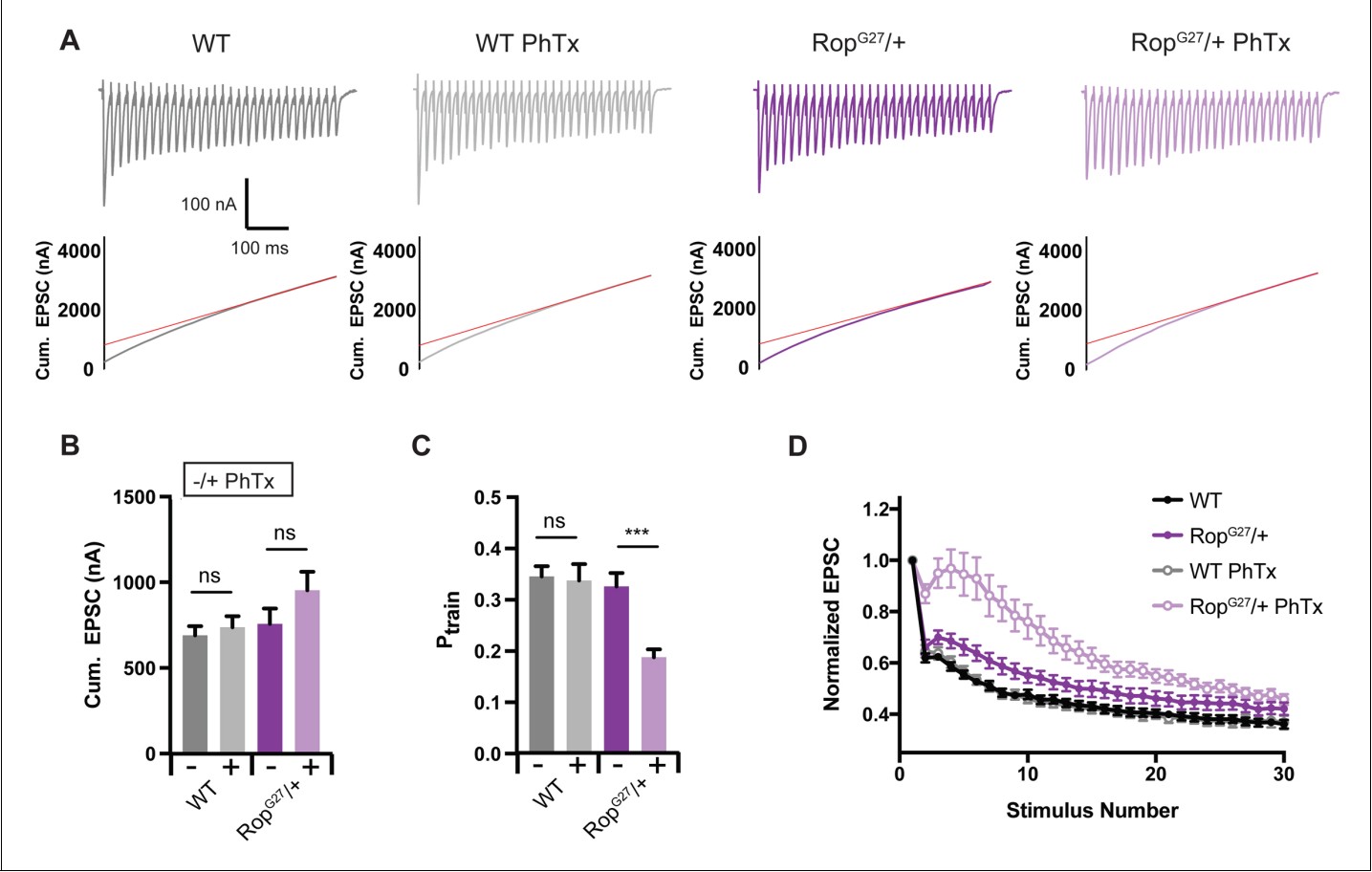

**Figure 5.** Decreased Release Probability in *Rop* Mutants. (**A**) Sample EPSC traces (top) and cumulative EPSC amplitudes (bottom) ±PhTx for indicated genotypes. Experiment used 60 Hz stimulation (30 stimuli) in 1.5 mM [$Ca^{2+}$]$_e$. Red line is fit to cumulative EPSC data and back extrapolated to time zero. (**B**) Average cumulative EPSC amplitudes ± PhTx for indicated genotypes; ns, not significant. Student's t-test, two tailed. (**C**) Average P$_{train}$ ±PhTx for indicated genotypes. P$_{train}$ = 1st EPSC/cumEPSC; ns, not significant; ***p<0.001. (**D**) Average EPSC amplitudes normalized to the first pulse are plotted against stimulus number for indicated genotypes.
DOI: https://doi.org/10.7554/eLife.40385.007

sensitive to *Rop* haplo-insufficiency and Rop is not limiting for the homeostatic potentiation of the RRP.

It remains apparent, however, that the initial EPSC of the stimulus train in *Rop^{G27}/+* mutants, recorded in the presence of PhTx, is smaller than that observed in wild type (*Figure 5A–B* see also *Figure 3C* for quantification). Thus, the homeostatic potentiation of the initial EPSC amplitude is disrupted in the *Rop^{G27}/+* mutants, whereas the homeostatic potentiation of the RRP is normal. This is quantified by dividing the initial EPSC amplitude by the cumulative EPSC amplitude during the stimulus train, a parameter referred to as P$_{train}$ (*Figure 5C*). It is apparent that release dynamics are altered as a consequence of decreased P$_{train}$ in the *Rop^{G27}/+* mutant in the presence of PhTx. Super-priming should contribute significantly to vesicle release in response to the first action potential of a stimulus train. Thus, one explanation for this result is that the super-primed vesicle pool has not been appropriately expanded in the *Rop^{G27}/+* mutant. During the stimulus train, elevated intra-terminal calcium could overcome *Rop^{G27}/+* haploinsufficiency by driving the normal priming process, leading to expansion of the RRP in the presence of PhTx. These data argue that Rop may be essential for the expansion of the super-primed vesicle pool during expression of PHP. It is worth noting that this is the first example of a significant alteration in presynaptic release dynamics that is specific to the induction of PHP.

## A genetic interaction of rop and RIM during PHP

There has been considerable progress identifying presynaptic proteins that are necessary for presynaptic homeostatic plasticity (*Davis, 2013*; *Müller et al., 2015*). These genes include the active zone associated scaffolding proteins RIM (Rab3 Interacting Molecule) and RBP (RIM Binding Protein), both of which are components of a proposed molecular priming pathway within the presynaptic nerve terminal (*Südhof, 2012a*). If Rop is integrally involved in the mechanisms of PHP within the presynaptic terminal, potentially acting in the priming process during PHP, then we might expect genetic interactions with *rim* and *RBP*.

Genetic interactions were performed by assaying heterozygous, null mutations alone and in comparison to the effects observed in a double heterozygous condition (*Frank et al., 2009*; *Müller et al., 2015*). We first assayed baseline release. At 0.3 mM $[Ca^{2+}]_e$, baseline release was normal in both $Rop^{G27}/+$ and $rim^{103}/+$ (*Figure 6B*). However, release (quantal content; 6C) was diminished by nearly 50% in the double heterozygous condition (WT QC = 34.8 ± 0.8; $Rop^{G27}/+$ QC = 30.1±1.6; $rim^{103}/+$ QC = 32.3±1.4; $Rop^{G27}/rim^{103}$ QC = 18.7 ± 1.3) (*Figure 6C*). This is a very strong genetic interaction for baseline release, even by comparison with previously published genetic interactions of other genes with *rim* (*Wang et al., 2016*; *Orr et al., 2017*; *Hauswirth et al., 2018*).

Next, we assessed PHP. Upon application of PhTx, a heterozygous null mutation in *rim* ($rim^{103}/+$) causes a suppression of PHP (*Figure 6D*). This suppression is similar in magnitude to that observed in the $Rop^{G27}/+$ heterozygous null mutation (*Figure 6D*). However, when we examine a double heterozygous mutant with $Rop^{G27}/+$ placed in trans to $rim^{103}/+$, PHP is completely blocked (*Figure 6C,D*). To underscore the robustness of this genetic interaction, we plot the relationship between mEPSP amplitude and quantal content for individual recordings (*Figure 6E*). Each data point represents the average mEPSP and quantal content for a single NMJ recording (see also *Hauswirth et al., 2018*). In wild type, data can be fit with a line representing the homeostatic process. We also present dotted lines that encompass 95% of all of the data points in the wild type graph. This fit and 95% data interval are used to compare data distributions to other mutant backgrounds. In both of the single heterozygous mutants ($Rop^{G27}/+$ and $rim^{103}/+$) the data points lie below the best-fit line for wild type, but are largely retained within the 95% interval, consistent with a minor suppression of PHP when all data points are averaged (*Figure 6C,D*). However, in the double heterozygous animal ($Rop^{G27}/rim^{103}$) it is clear that PHP fails and the majority of data points in the presence of PhTx reside outside the interval that contains 95% of all wild type data points. More specifically, for heterozygous null *Rop* and *rim* recordings in the presence of PhTx, 25% and 16% of these individual data points in the presence of PhTx fall outside of the lines encompassing 95% of wild-type data (*Figure 6E*). In the double heterozygous mutants, 87% of PhTx data fall outside of the 95% confidence intervals (*Figure 6E*). This genetic interaction, referring specifically to PHP expression, is also conserved at elevated calcium levels (1.5 and 3.0 mM calcium; data not shown).

At baseline, there is a strong synergistic interaction between Rop and RIM, suggesting that both are functioning to control the same process relevant to vesicle release, perhaps synaptic vesicle priming (*Gulyás-Kovács et al., 2007*; *Koushika et al., 2001*). It remains unclear whether *rop* and *rim* function in the same genetic pathway based on this genetic interaction. The data are equally compatible with parallel pathways converging on the mechanism of PHP. Regardless, our data underscore that Rop has an essential function during the process of PHP. We consider this a particularly important line of reasoning, since it is not feasible to eliminate *Rop* and assess a block in PHP.

Next, we sought to understand the specificity of the genetic interaction between *Rop* and *rim*. To do so, we performed a series of additional genetic interactions with other genes previously implicated in the presynaptic synaptic vesicle priming process. First, we asked whether *Rop* shows a genetic interaction with *RIM Binding Protein (RBP)*. RBP biochemically interacts with RIM and is thought to form an extended presynaptic scaffold (*Wang et al., 2000*; *Liu et al., 2011*). In *Drosophila*, RBP is also essential for both baseline release and PHP (*Müller et al., 2015*). But, the mechanism by which RBP participates in PHP is distinct from the mechanism by which RIM participates in PHP (*Müller et al., 2015*). The heterozygous null mutation in *rbp/+* has no effect on baseline transmission at 0.3 mM external calcium, consistent with prior observations (*Müller et al., 2015*) (*Supplementary file 1*). We then demonstrate that PHP is also normal in the $Rop^{G27}/rbpSTOP^{STOP1}$ double heterozygous mutant (*Figure 6F*). This is evidence that there is specificity to the *rop*

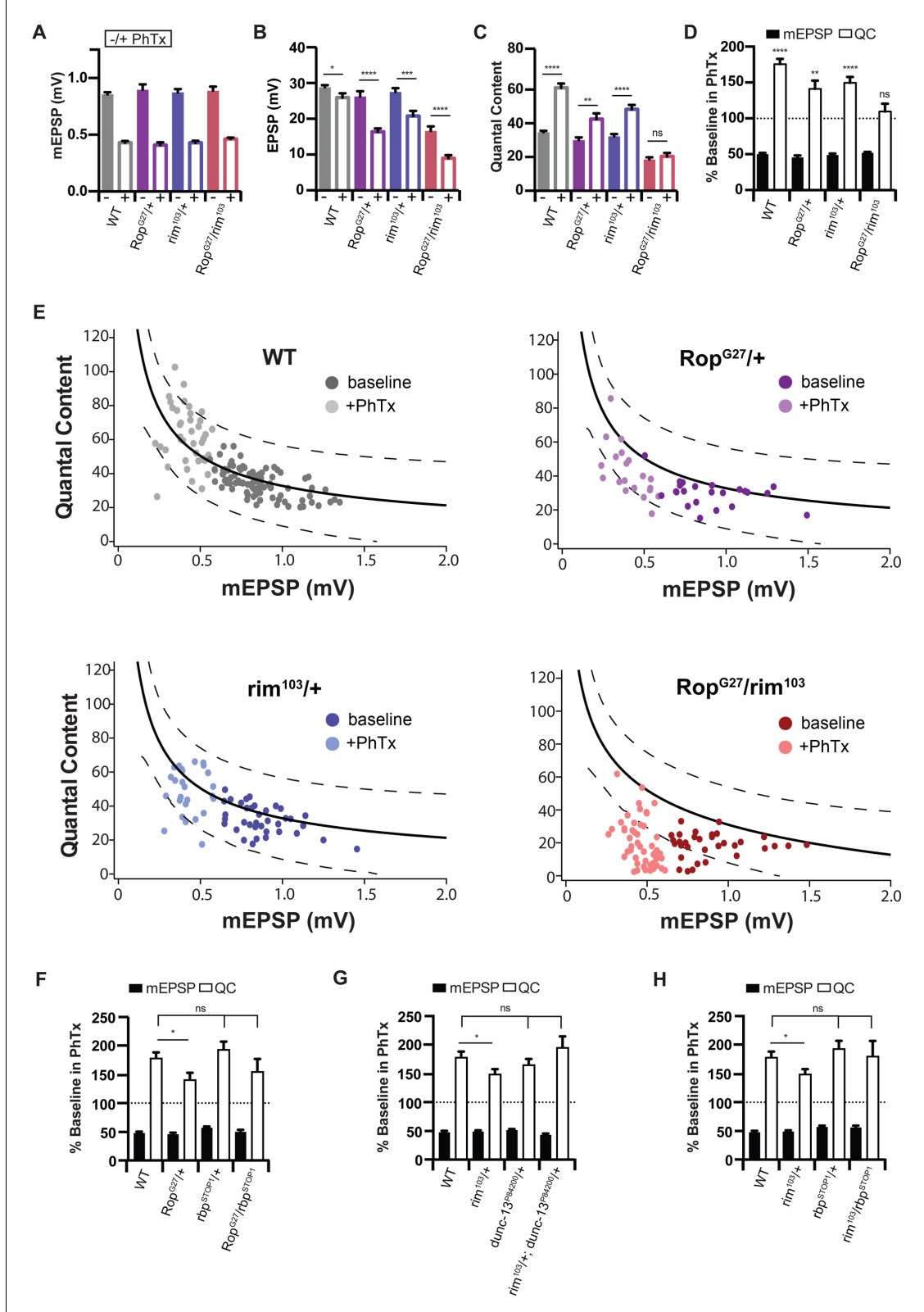

**Figure 6.** *Rop* Interacts with *rim* during Synaptic Homeostasis. (**A**) Average data for mEPSP amplitude ±PhTx for WT, heterozygous *Rop^G27^/+* mutant heterozygous *rim^103^/+* mutant, and transheterozygous *Rop^G27^/rim^103^* mutant at 0.3 mM [Ca²⁺]ₑ. PhTx application reduces amplitude in all genotypes (p<0.01). Data represent mean ±SEM. Student's t test. (**B**) Average data for EPSP amplitude ±PhTx for indicated genotypes; statistics as in (**A**); *p<0.05; ***p<0.001; ****p<0.0001, Student's t-test, two tailed. (**C**) Average data for Quantal Content ±PhTx for indicated genotypes as in (**A**); ns, not significant;

*Figure 6 continued on next page*

*Figure 6 continued*

**p<0.01; ****p<0.0001; Student's t-test, two tailed. (D) Average data for mEPSP and quantal content normalized to values in the absence of PhTx for indicated genotypes. Statistical comparisons are made within each genotype to baseline in the absence of PhTx. Student's t-test, two tailed. (E) Each point represents average data from an individual NMJ recording. For WT, recordings in the absence of PhTx are dark gray, those with PhTx are light gray. For $Rop^{G27}$/+, recordings in the absence of PhTx are dark purple, those with PhTx are light purple. For $rim^{103}$/+, recordings in the absence of PhTx are dark blue, those with PhTx are light blue. For $Rop^{G27}$/$rim^{103}$, recordings in the absence of PhTx are dark red, those with PhTx are light red. The black line in the WT graph is a curve fit to this control data. The same wild type curve-fit is overlaid on all other genotypes for purposes of comparison. Dotted black lines encompass 95% of wild type data points. These same lines from wild type are superimposed on the graphs for indicated genotypes. (F–H) Average percent change in mEPSP amplitude and quantal content in PhTx compared to baseline for trans heterozygous combinations: $Rop^{G27}$/$rbp^{STOP1}$ (F), $rim^{103}$/+; $dunc13^{P84200}$/+ (G), $rim^{103}$/$rbp^{STOP1}$ (H); ns, not significant; *p<0.05; Student's t-test, two tailed.
DOI: https://doi.org/10.7554/eLife.40385.008

interaction with *rim* during PHP. It is worth noting that PHP is highly variable in the $Rop^{G27}$/$rbpSTOP^{STOP1}$ double heterozygous mutant, preventing us from making any additional conclusions regarding whether PHP is similar to, or different from wild type.

Next, we asked whether *rim* genetically interacts with *Unc-13* during baseline neurotransmitter release and PHP. Munc-13 is known to biochemically interact with RIM (*Betz et al., 2001*). Alone, the *dunc-13$^{P84200}$*/+ heterozygous null mutation has no effect on baseline transmission or PHP (*Supplementary file 1*). Remarkably, the double heterozygous condition of *rim* and *dunc-13* also has no effect on baseline transmission or PHP (*Supplementary file 1* and *Figure 6G*). A similar set of findings is observed when we tested a double heterozygous condition of *rim* with *rbp* (*Supplementary file 1* and *Figure 6H*). While baseline transmission is decreased in the double heterozygous combination of *rim*/+ and *rbp*/+, we find that PHP is normal, confirming previously published data (*Müller et al., 2015*). It is somewhat surprising that the *rim*/+; *unc13*/+ double heterozygous mutant has significantly more PHP than observed in *rim*/+ alone, perhaps relating to the functional importance of the RIM-Unc13 biochemical interaction during PHP. Regardless, when taken together, our results underscore the specificity and importance of the genetic interaction between *rop* and *rim* and the relevance of *rop* to the mechanisms of PHP.

## Loss of rop and rim limits the super-primed vesicle pool

We note that heterozygous $Rop^{G27}$/+ mutants have a defect in PHP that is apparent on the first action potential of a stimulus train, but the homeostatic expansion of the RRP is apparent upon further stimulation (*Figure 5*). Both Rop and Rim are well-established molecular players that contribute to synaptic vesicle priming (*Südhof, 2012a*; *Cook et al., 2012*). We returned to the use of PdBu to assess whether the block of PHP in the $Rop^{G27}$/$rim^{103}$ double heterozygous mutant is correlated with a deficit in vesicle super-priming. As outlined above, the magnitude of PdBu-dependent potentiation in response to a single action potential should reflect the balance of primed to super-primed vesicles. A large PdBu effect argues for a smaller pool of super-primed vesicle pool. We compared wild type to the $Rop^{G27}$/$rim^{103}$ double heterozygous mutant, recording in the presence and absence of PdBu (*Figure 7*). Experiments were performed at 0.75 mM $[Ca^{2+}]_e$, a condition in which PdBu potentiates wild type synapses by ~165% (*Figure 7A–D*). Next, we demonstrate that application of PdBu to each heterozygous mutant alone, either $Rop^{G27}$/+ or $rim^{103}$/+, is identical to wild type (*Figure 7C,D*; p>0.1 ANOVA). Then, we demonstrate that PdBu has a dramatically increased effect size when applied to the $Rop^{G27}$/$rim^{103}$ double heterozygous mutant, potentiating release by more than 350%, a dramatically increased effect size compared to application of PdBu to wild type and single heterozygous controls (*Figure 7A–D*, p<0.01; ANOVA). These data argue that the $Rop^{G27}$/$rim^{103}$ double heterozygous mutant limits the size of the super-primed vesicle pool, thereby impairing vesicle release in response to a single action potential, an effect that is strongly correlated with a block in the expression of PHP. Two other observations should be noted. First, the $Rop^{G27}$/$rim^{103}$ double heterozygous mutant recorded at 0.75 mM $[Ca^{2+}]_e$ has a greater effect on baseline release than observed at 0.3 mM $[Ca^{2+}]_e$. The basis for this effect is unknown. Second, we show that the enhanced effects of PdBu on the $Rop^{G27}$/$rim^{103}$ double heterozygous mutant is consistent with the potentiation of release probability and an associated change in paired pulse ratio (*Figure 7C*). This simply confirms that PdBu is behaving as expected when applied to the $Rop^{G27}$/$rim^{103}$ double heterozygous mutant. Again, this type of synergistic genetic interaction is generally taken as evidence

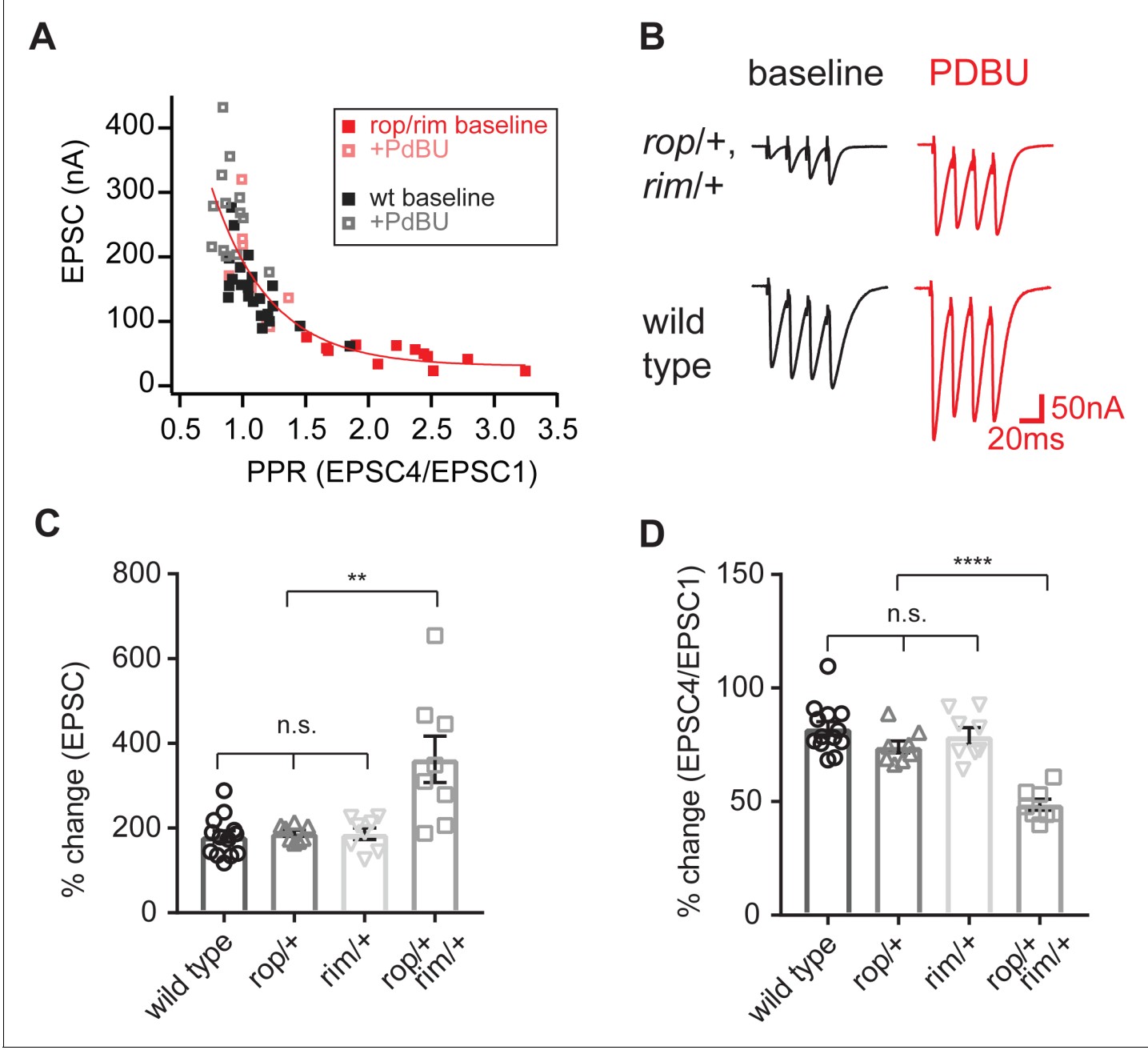

**Figure 7.** Enhanced action of PdBu at synapses depleted of both Rop and RIM. (**A**) Paired-pulse ratio is plotted against initial EPSC amplitude for the indicated genotypes and conditions (in the absence and presence of PdBu). (**B**) Representative traces for the indicated genotypes in the absence (baseline) and presence of PdBu (PdBu, red). (**C**) The effect of PdBu application is plotted for wild type controls, the *Rop^G27^/rim^103^* double heterozygous condition and each heterozygous mutation alone. Each genotype is expressed as a percent change in the presence compared to absence of PdBu. Calculations are made on the first EPSC of the stimulus train (EPSC1). The average percent change is statistically significant only in the double heterozygous condition (p<0.01; ANOVA, One Way with Tukey Multiple Comparisons). (**D**) Data as in (**C**) plotting the ratio of EPSC4/EPSC1 as a percent change in the presence compared to absence of PdBu. The average percent change is statistically significant only in the double heterozygous condition (p<0.01; ANOVA, One Way with Tukey Multiple Comparisons).

DOI: https://doi.org/10.7554/eLife.40385.009

that two genes participate in the same process. It is not possible to make any conclusion regarding whether these genes are acting in a linear signaling pathway or in parallel signaling pathways. But, based on formal genetic argument, we are able to conclude that they likely converge to control the same processes at the presynaptic terminal, both the PdBu sensitive vesicle pool and PHP.

## Loss of *syx1A* rescues PHP in the heterozygous *rop* mutant background

Unc-18 is a well-established syntaxin binding protein (*Hata et al., 1993*; *Pevsner et al., 1994*; *Halachmi et al., 1995*). The Unc-18 interaction with Syntaxin is complex, including progressive interactions with closed and open conformations of Syntaxin. Ultimately, the Unc-18 interaction with open Syntaxin is thought to catalyze SNARE assembly, greatly decreasing the energy barrier to calcium-driven vesicle fusion. Indeed, Unc-18 binding to open Syntaxin is believed to be a prerequisite for efficient synaptic vesicle fusion (*Dulubova et al., 2007*; *Deák et al., 2009*). Based on the genetic interactions reported above, we expected that loss of *syx1A* would strongly enhance the loss of function phenotype of the heterozygous $Rop^{G27}/+$ mutant, resulting in diminished vesicle release and a block of PHP.

We acquired a previously published null mutation in *syntaxin1A* and examined baseline neurotransmitter release in a heterozygous *syx1A* mutant ($syx1A^{\Delta229}$; *Schulze et al., 1995*). There is no change in baseline mEPSP amplitude, EPSP amplitude or quantal content in $syx1A^{\Delta229}/+$ assayed at 0.3 mM $[Ca^{2+}]_e$ (*Figure 8A–D*). Thus, *syntaxin* is not haplo-insufficient for baseline release. Remarkably, the same is true for the double heterozygous condition, combining $syx1A^{\Delta229}/+$ with $Rop^{G27}/+$. Baseline transmission is normal compared to wild type (*Figure 8D*). In other systems, it is estimated that Syntaxin1A is present in ~5 fold excess compared to the levels of synaptic Unc-18 (*Graham et al., 2004*). This could explain the lack of a genetic interaction at baseline.

Next, we examined PHP. A heterozygous null mutation in *syx1A* ($syx1A^{\Delta229}/+$) has normal PHP (*Figure 8E*), again consistent with a possible excess of Syntaxin protein at the release site. The heterozygous null mutation in *Rop* ($Rop^{G27}/+$) suppresses homeostatic potentiation (*Figure 8E*), confirming experiments presented earlier in this study. Remarkably, when we place $Rop^{G27}$ in trans to $syx1A^{\Delta229}$ (double heterozygous condition), we find that homeostasis is fully expressed (*Figure 8E*). Impaired PHP caused by the $Rop^{G27}/+$ mutation is completely rescued to wild type levels. More specifically, in the double heterozygous mutants, EPSP amplitudes fully compensate in the presence of PhTx (*Figure 8C*) and there is a wild type level enhancement of quantal content (*Figure 8E*) in the presence of PhTx. Finally, the rescue of the $Rop^{G27}/+$ mutation by $syx1A^{\Delta229}/+$ is not restricted to PHP. The rate of spontaneous vesicle fusion is also restored to wild type levels, underscoring the validity of this genetic rescue (*Figure 8F*). Note that there is, as yet, no indication that mEPSP frequency is directly responsible for the induction of PHP (*Frank et al., 2006*; *Frank et al., 2009*; *Harris et al., 2015*; *Goold and Davis, 2007*). Since we are examining heterozygous, null mutations, the most parsimonious conclusion is that Syntaxin normally functions to restrict the action of Rop that is required for PHP. Reducing the level of Syntaxin relieves a restriction on Rop activity and restores full expression of PHP. To provide further evidence for this surprising finding, we sought to disrupt the physical interaction of Rop and Syntaxin and assess whether this might also rescue the expression of PHP in the $Rop^{G27}/+$ mutant background.

## Evidence that Syx1A restrains rop from participating in PHP

We performed an in vitro binding assay to confirm the biochemical interaction between Rop and Syntaxin1A. Recombinant wild-type Rop protein binds strongly to recombinant Syntaxin1A (GST-$syx1A^{\Delta C}$) (*Figure 9B–C*) ($K_D$ = 0.4 μM). Next, we surveyed previously characterized point mutations in the *Rop* gene, searching for candidate mutations that reside near the conserved Syntaxin binding interface (*Figure 9A*). Unc-18 binds to Syntaxin at two sites, an N-terminal region that interacts with the N-terminal peptide of Syntaxin and a helical region that represents a larger interaction surface. The $Rop^{G11}$ mutant harbors a point mutation (Asp 45→ Asn; *Harrison et al., 1994*) that resides within or directly adjacent to a predicted helical Syntaxin binding interface (*Misura et al., 2000*). Here, we demonstrate that this mutation completely abolishes in vitro binding between recombinant $Rop^{G11}$ mutant protein and recombinant GST-$syx1A^{\Delta C}$ protein (*Figure 9B–C*) ($K_D$ = 36.1 μM).

We next assayed baseline release in the $Rop^{G11}$ mutant, placed in trans to either a deficiency that removes the *Rop* gene locus, or the $Rop^{G27}$ null allele. To our surprise, both allelic combinations are

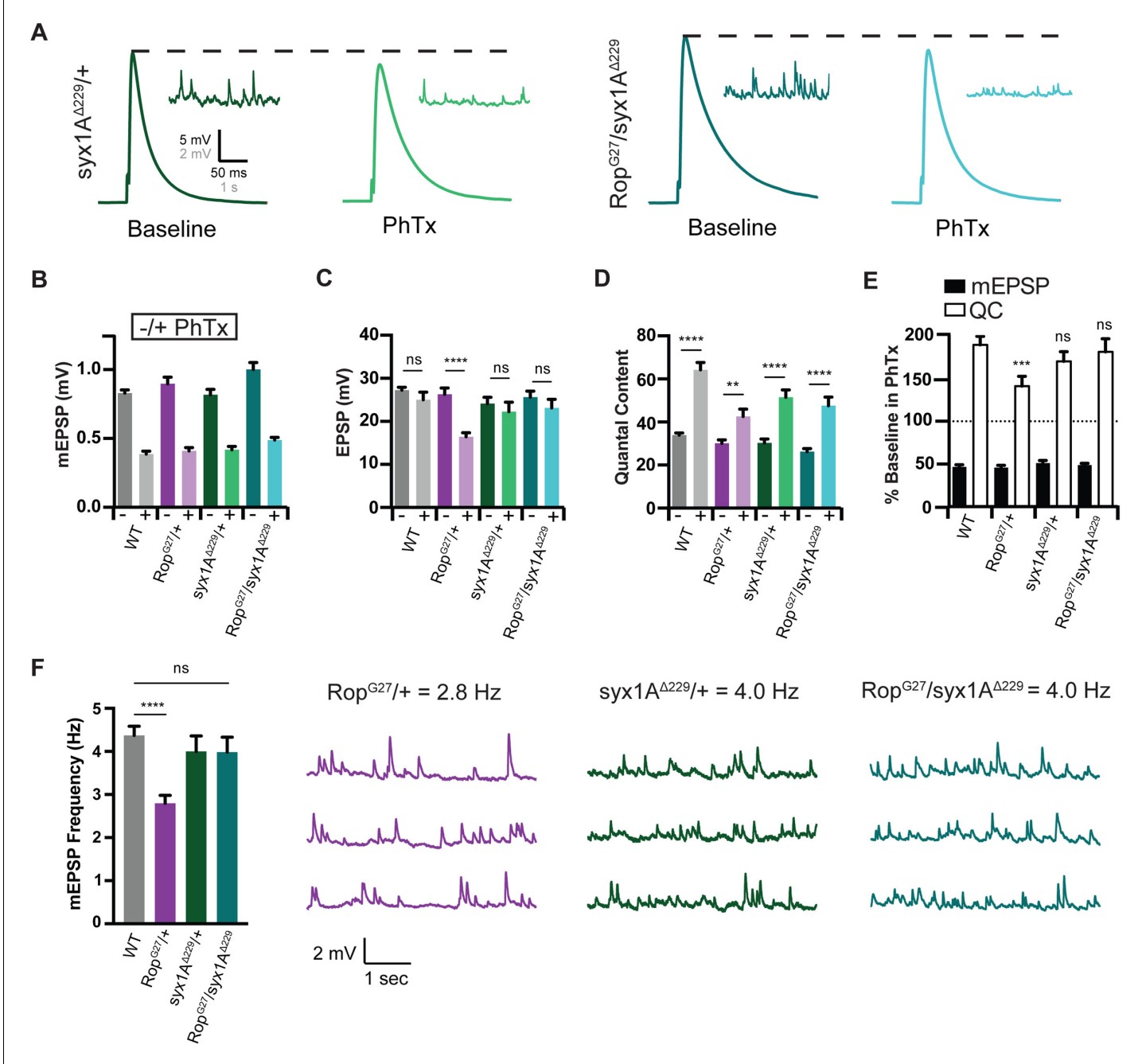

**Figure 8.** *Syx* rescues PHP during Synaptic Homeostasis. (A) Sample traces showing EPSP and mEPSP amplitudes in the absence (baseline) and presence (PhTx) of PhTx for *syx1A* heterozygous null allele (*syx1A^Δ229^/+*) and heterozygous *syx1A^Δ229^* placed in trans with *Rop^G27^* mutant (*Rop^G27^/syx1A^Δ229^*). (B) Average data for mEPSP amplitude ±PhTx for indicated genotypes. PhTx application reduces amplitude in all genotypes (p<0.01). Data represent mean ±SEM. Student's t test. (C–E) Average data for EPSP amplitude (C) Quantal Content (D) and mEPSP and quantal content normalized to values in the absence of PhTx (E) for indicated genotypes. Statistical comparisons are made to wild type for each genotype in (E). **p<0.01; ****p<0.0001, Student's t-test, two tailed. (F) Average mEPSP frequencies (Hz) (left) and sample traces showing mEPSPs for indicated genotypes (right). Student's t test, two tailed.

DOI: https://doi.org/10.7554/eLife.40385.010

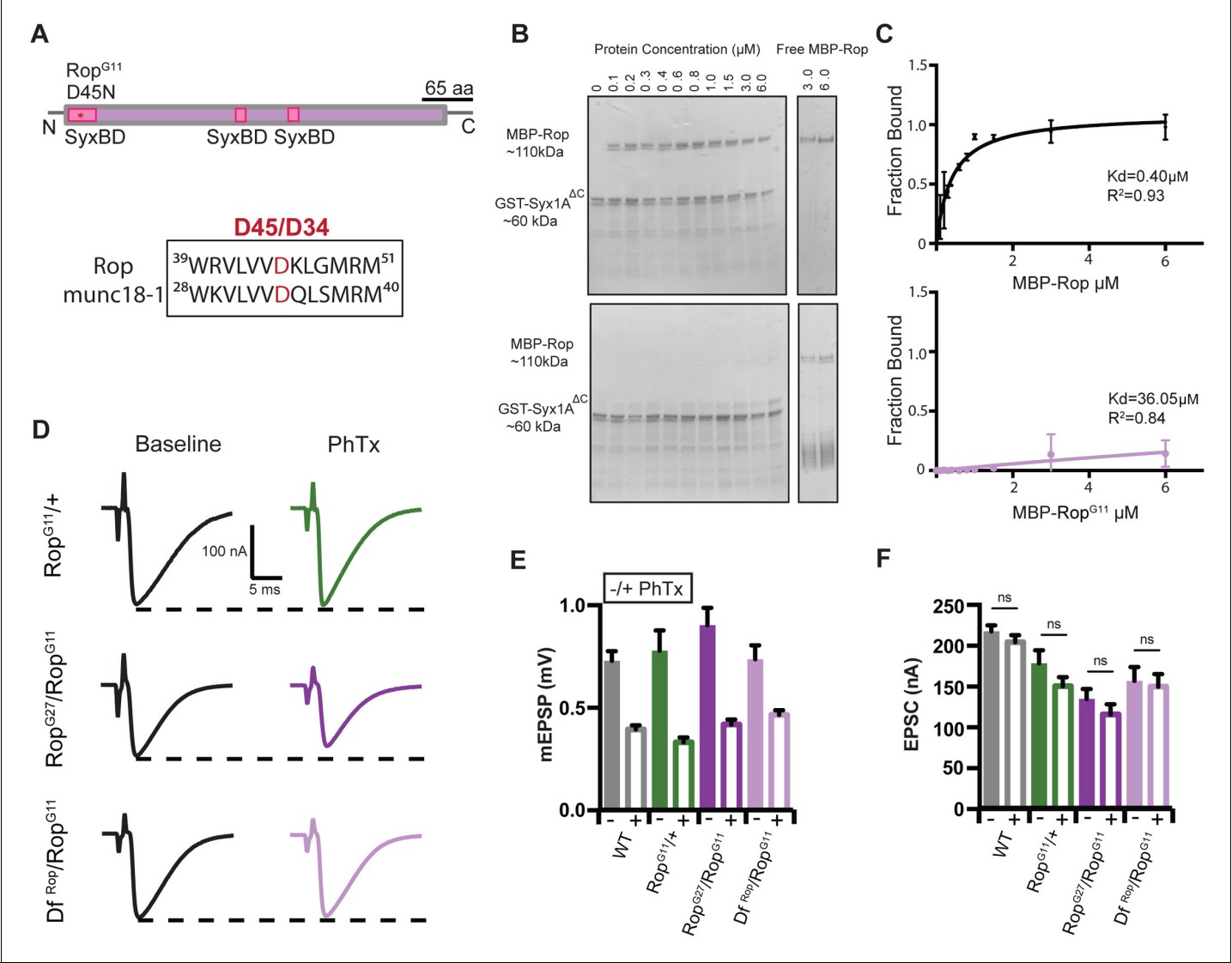

**Figure 9.** *Rop^{G11}* abolishes the biochemical interaction between Rop and syx1A and rescues homeostasis defect in Rop. (**A**) Schematic of the *Drosophila* Rop protein. Point mutation *Rop^{G11}* is indicated by red star at syntaxin-binding domain (SBD) of the Rop protein shown in pink. *Rop^{G11}* converts Aspartic Acid (D45) to Asparagine (N). This site is conserved in mammalian *Rop (munc18-1)*. (**B**) Coomassie stains of in vitro binding assays. (left) MBP-Rop (110 kDa) coprecipitated with bead-bound GST fusions of syx1A (GST-syx1A^{ΔC}) (60 kDa). MBP-Rop does not bind to GST-syx1A^{ΔC} in the presence of single point mutation at the N-terminal of Rop (MBP-Rop^{G11}). (right) Free MBP-Rop in the absence of GST-syx1A. (**C**) Binding curves quantify dissociation constant (K_d) for MBP-Rop and MBP-Rop^{G11} binding to GST-syx1A^{ΔC}; x-axis is concentration of MBP recombinant protein used (μM); y-axis is the fraction of protein bound; n = 2 (**D**) Sample traces showing EPSC amplitudes ± PhTx for *Rop^{G11}* heterozygous null allele (*Rop^{G11}/+*), heterozygous *Rop^{G11}* placed in trans with *Rop^{G27}* mutant (*Rop^{G27}/RopG^{G11}*), and heterozygous *Rop^{G11}* placed in trans with *Df^{Rop}* (*Df^{Rop}/RopG^{G11}*). (**E**) Average data for mEPSP amplitude ±PhTx for indicated genotypes. PhTx application reduces amplitude in all genotypes (p<0.01). Data represent mean ±SEM. Student's t test. two tailed. (**F**) Average data for EPSC amplitude as in (**B**); ns, not significant; Student's t test, two tailed.
DOI: https://doi.org/10.7554/eLife.40385.011

viable to the third instar stage and we observe robust neurotransmitter release (*Figure 9D*). Specifically, we find that EPSC amplitudes are decreased by ~45% on average in both the *Rop^{G11}/Df^{Rop}* and *Rop^{G11}/Rop^{G27}* allelic combinations compared to wild type (*WT* EPSC = 217.4 nA±7.3; *Rop^{G27}/Rop^{G11}* EPSC = 134.2 nA±12.2; p<0.01; *Df^{Rop}/Rop^{G11}* EPSC = 156.3 nA±17.2; p<0.01) (*Figure 9F*). This effect is more severe than the defect observed in the *Rop^{G11}/+* heterozygous condition, consistent with loss of Rop function in both the *Rop^{G11}/Df^{Rop}* and *Rop^{G11}/Rop^{G27}* allelic combinations. But, the presence of synchronous release is very surprising, since there is no wild type Rop protein

at the synapse. From these data we conclude that the $Rop^{G11}$ mutation does not completely block all interactions between Rop and Syntaxin in vivo. The in vitro binding assay is predicted to test the binding of Rop to the closed conformation of Syntaxin. We speculate, based on work in other systems, that $Rop^{G11}$ could be localized to the SNARE complex through other molecular interactions, in vivo. Once at the release site, $Rop^{G11}$ mutant protein might still interact with the open conformation of Syntaxin and facilitate SNARE-mediated fusion.

Next, we assayed PHP in the $Rop^{G11}/+$ as well as the $Rop^{G11}/Df^{Rop}$ and $Rop^{G11}/Rop^{G27}$ allelic combinations. In the presence of PhTx, EPSC amplitudes are restored to baseline values in the $Rop^{G11}/+$ mutant, indicative of fully functional PHP (*Figure 9D–F*). Remarkably, PHP is also fully expressed in both the $Rop^{G11}/Df^{Rop}$ and $Rop^{G11}/Rop^{G27}$ allelic combinations (*Figure 9D–F*). Thus, the presence of the $Rop^{G11}$ allele fully rescues PHP in the presence of the $Rop^{G27}/+$ as well as the $Df^{Rop}/+$ alleles. While surprising, these data are entirely consistent with the observation that a heterozygous null mutation in *syntaxin1A* rescues PHP in the $Rop^{G27}$ mutant background (*Figure 8*). Taken together, our data are consistent with an emerging model in which Syx1A regulates the availability of Rop to participate in PHP (see discussion).

## Discussion

We have advanced our understanding of presynaptic homeostatic plasticity in several important ways. First, we demonstrate that *rop* (*Unc-18*) is essential for PHP, significantly extending prior evidence presented at the mouse NMJ (*Sons et al., 2003*). Thus, we argue that *rop* (*Unc-18*) is an evolutionarily conserved component of PHP signaling, enabling expression of PHP at the NMJ of Drosophila (this paper) and mice (*Sons et al., 2003*). Indeed, Unc-18 is the first molecular signaling component demonstrated to have a conserved, required function during PHP in both systems. Fly and mouse are separated by nearly 500 million years of evolution, suggesting that the molecular mechanisms of PHP may be as ancient as mechanisms that achieve action-potential induced, calcium-dependent, neurotransmitter release.

Second, our data demonstrate that Rop is limiting for the expression of PHP. Prior work at the mouse NMJ was the first to provide evidence that Unc-18 might participate in PHP (*Sons et al., 2003*). However, the prior work only included an analysis of *Unc-18/+* heterozygous null mutant animals and, ultimately, could not rule out the formal possibility that Unc-18 was essential for baseline release and, as a secondary consequence, limited the expression of PHP. We provide several lines of evidence supporting the conclusion that Unc-18 has an activity that is necessary for PHP. For example, neurotransmitter release in the heterozygous $Rop^{G27}/+$ mutant remains highly sensitive to changes in extracellular calcium. Yet, across a 10-fold range of extracellular calcium, PHP is suppressed by a constant fraction of ~30%. We also pursued a series of genetic interactions and provide evidence for a strong, specific, genetic interaction of *Rop* with *rim*, placing Unc-18 within a known PHP signaling framework.

Third, Unc-18 is the first integral component of the synaptic vesicle fusion apparatus to be linked to the expression of PHP. As such, our data provide a reasonable endpoint for the presynaptic homeostatic signaling system. In recent years, trans-synaptic signaling molecules have been shown to be required for PHP including Semaphorin/Plexin signaling (*Orr et al., 2017*), innate immune signaling (*Harris et al., 2015*) and signaling from the synaptic matrix (*Wang et al., 2016*). Many of these signaling systems interact with the presynaptic scaffolding protein RIM (*Harris et al., 2015*; *Wang et al., 2016*; *Orr et al., 2017*; *Hauswirth et al., 2018*). But, it was previously unknown what molecular mechanisms reside downstream of RIM, since RIM is a molecular scaffolding protein. We cannot formally conclude, based on our genetic data, that Rop functions downstream of RIM. However, it is clear that Rop is in a position to directly modulate the fusion apparatus and, as such, is very likely to mediate signaling that is localized to the active zone by the RIM-dependent cytomatrix.

The rescue of PHP by loss of Syntaxin or by disruption of Rop-Syntaxin binding is a surprise, but one that can be understood when placed in the context of work previously documented in other systems. A recent single molecule imaging study demonstrates that the majority of Unc-18 may reside outside of active zone with limited mobility (*Smyth et al., 2013*). It has also been shown that Syntaxin is present in large excess compared to Unc-18, and Syntaxin protein is broadly distributed beyond sites of synaptic vesicle fusion (*Broadie et al., 1995*; *Graham et al., 2004*). Thus, the majority of Unc-18 protein could interact with Syntaxin in the peri-active zone, presumably binding a

closed Syntaxin conformation. Accordingly, Unc-18 would be in equilibrium, moving between a peri-active zone reservoir and fusion competent vesicles at the active zone. In this way, Syntaxin could restrict the amount of Unc18 available for participation in synaptic vesicle fusion at the release site. Thus, when we remove one copy of the *syntaxin* gene, or diminish the binding of Unc-18 to the closed confirmation of Syntaxin, we might be shifting the distribution of Unc-18 toward the release site and achieve a rescue PHP. This model is consistent with data in other systems, demonstrating that the levels of Unc18 at the release site can influence vesicle release rate. For example, over-expression of *unc-18* in mice is sufficient to potentiate vesicle release (*Voets et al., 2001*), albeit to a limited extent (*Toonen and Verhage, 2003*). Thus, we propose that the expression of PHP involves an as yet unknown signaling event that mobilizes Unc-18 to the active zone where it is sufficient to promote vesicle priming as a final stage necessary for the full expression of PHP.

## PHP potentiates release without altering release dynamics

Homeostatic signaling systems are powerful corrective processes. One of the most remarkable properties of PHP is that synaptic gain is controlled without altering presynaptic release dynamics (*Figure 1*). The expression of PHP includes a required potentiation of presynaptic calcium influx (*Müller and Davis, 2012a*), a required expansion of the readily releasable synaptic vesicle pool (*Müller et al., 2012b*; *Müller et al., 2015*) and, as demonstrated here, increased function of Unc-18 dependent vesicle priming. Only when all three processes are simultaneously potentiated is it possible to achieve increased presynaptic release while precisely preserving presynaptic release dynamics. The capacity to sustain release dynamics underscores the emerging molecular complexity of PHP signaling. Each of the three processes that control release are under homeostatic control. A PHP-dependent change in calcium influx is controlled by ENaC channel insertion (*Younger et al., 2013*). The PHP-dependent modulation of the RRP involves signals converging on the synaptic cytomatrix including RIM, RBP and regulation of presynaptic actin (*Müller et al., 2012b*; *Müller et al., 2015*; *Orr et al., 2017*). Future work will be necessary to establish the inter-dependence of these presynaptic homeostatic signaling pathways.

Here, we present evidence that one function of Unc18 may be to ensure that the ratio of primed to super-primed vesicles remains constant during the doubling of presynaptic release occurring during PHP. Several lines of evidence support this conclusion. PHP is fully expressed following single action potential stimulation, necessitating expansion of the docked/primed vesicle pool. We use PdBu to determine the fraction of docked/primed vesicles that reside in the super-primed state versus a lower-release probability docked/primed stated. We find that the ratio of primed to super-primed vesicles remains constant during PHP. When PdBu is applied to the NMJ, converting the entire pool to a super-primed state, release dynamics are converted to synaptic depression, arguing that preservation of the primed to super-primed ratio is essential for maintaining wild type release dynamics. Then, two results connect Unc18 function to control of the super-primed pool. First, loss of Rop (Unc18) primarily affects the first EPSC of a stimulus train, consistent with control of the super-primed vesicle pool. Second, when $Rop^{G27}$/+ is combined with $rim^{103}$/+, the double heterozygous condition completely blocks PHP expression and there is a dramatic loss of the super-primed population of vesicles, as revealed by a ~ 300% increase in the effect of PdBu on presynaptic release.

The preservation of presynaptic release dynamics during PHP seems to be a fundamental property of PHP. Not only is synaptic gain stabilized, but also the dynamic, activity-dependent transfer of information at a synapse is precisely preserved. At the neuromuscular junction, the impact is presumably to maintain the quality of muscle excitation. If extended to the central nervous system, where PHP is also observed (*Davis, 2013*), preservation of release dynamics would stabilize the flow of information through complex neural circuitry, with obvious relevance to processes such as sensory-motor integration and other neural computations. Clearly, many processes must be coordinately controlled by the intracellular signaling systems that participate in presynaptic homeostatic plasticity in order to double vesicular release, at a constant number of active zones, while precisely preserving presynaptic release dynamics. Here, we identify a novel mechanism that participates in PHP and the preservation of presynaptic release dynamics, the regulated action of Unc18 at the presynaptic release site. This is an important advance toward what must become a systems biology level solution to regulation of presynaptic neurotransmitter release during PHP at central and peripheral synapses.

# Materials and methods

## Key resources table

| Reagent type (species) or resource | Designation | Source or reference | Identifiers | Additional information |
|---|---|---|---|---|
| Gene (*Drosophila melanogaster*) | *Rop* | NA | FLYB: FBgn0004574 | |
| Gene (*D. melanogaster*) | *Rim* | NA | FLYB: FBgn0053547 | |
| Gene (*D. melanogaster*) | *Rbp* | NA | FLYB: FBgn0262483 | |
| Gene (*D. melanogaster*) | *unc-13* | NA | FLYB: FBgn0025726 | |
| Gene (*D. melanogaster*) | *Syx1A* | NA | FLYB: FBgn0013343 | |
| Strain - strain background | WT - $w^{1118}$ | NA | $w^{1118}$ | |
| Genetic reagent (*D. melanogaster*) | $Rop^{G27}$ | Bloomington Drosophila Stock Center | BDSC: 4381 FLYB FBst0 004381 RRID: DGGR_107715 | Flybase symbol: bw (*Aravamudan et al., 1999*); Rop[G27] st (*Aravamudan et al., 1999*) /TM6B, Tb[+] |
| Genetic reagent (*D. melanogaster*) | $Df^{Rop}$ (Df(3L) BSC735) | Bloomington Drosophila Stock Center | BDSC: 26833 FLYB FBst0026833 RRID: BDSC_26833 | Flybase symbol: w[1118]; Df(3L) BSC735/ TM6C, Sb (*Aravamudan et al., 1999*) cu (*Aravamudan et al., 1999*) |
| Genetic reagent (*D. melanogaster*) | $elav^{C155}$-GAL4 | Bloomington Drosophila Stock Center | BDSC: 458 FLYB FBst0000458 RRID: BDSC_458 | Flybase symbol: P{w[+mW.hs]=GawB}elav [C155] |
| Genetic reagent (*D. melanogaster*) | *UAS-Rop RNAi* | Vienna Drosophila RNAi Center | VDRC: 19696 FLYB FBst0 453580 RRID: FlyBase _FBst0453580 | Flybase symbol: w[1118]; P{GD1523 }v19696/TM3 |
| Genetic reagent (*D. melanogaster*) | $rim^{103}$; rim | (*Müller et al., 2012b*) PMID: 23175813 | | |
| Genetic reagent (*D. melanogaster*) | $rbp^{STOP1}$ | (*Liu et al., 2011*) PMID: 22174254 | | gift from Stephan Sigrist |
| Genetic reagent (*D. melanogaster*) | $dunc-13^{P84200}$ | Kyoto Stock Center | KSC: 101911 RRID: DGGR_101911 | Flybase symbol: ry[506]; P{ry11} l(4)ry16 (*Aravamudan et al., 1999*) /ci[D] |
| Genetic reagent (*D. melanogaster*) | $syx1A^{\Delta229}$ | Kyoto Stock Center | KSC: 107713 RRID: DGGR_107713 | Flybase symbol: Syx1A[Delta229] ry[506]/TM3, ry[RK] Sb(*Aravamudan et al., 1999*) Ser(*Aravamudan et al., 1999*) |
| Genetic reagent (*D. melanogaster*) | $Rop^{G11}$ | (*Harrison et al., 1994*) PMID: 7917291 | | gift from Hugo Bellen |

*Continued on next page*

*Continued*

| Reagent type (species) or resource | Designation | Source or reference | Identifiers | Additional information |
|---|---|---|---|---|
| Recombinant DNA reagent | PMAL-c5E (vector) | New England Biolabs | NEB: N8110 | |
| Recombinant DNA reagent | PGEX-4T1 (vector) | Addgene | 27-4580-01 | |
| Recombinant DNA reagent | Rop (cDNA) | Drosophila Genomics Resource Center | DGRC: SD04216 | |
| Recombinant DNA reagent | Syx1A (cDNA) | Drosophila Genomics Resource Center | DGRC: LD43943 | |
| Recombinant DNA reagent | PMAL-Rop$^{WT}$ (plasmid) | This paper | | Primers CGCGGATC CATGGCCTTG AAAGTGCTGGTG G and CC GGAATTCTTA GTCCTCC TTCGAGAGACTGC were used to amplify Rop, which was then cloned into PMAL-5ce vector |
| Recombinant DNA reagent | PMAL-Rop$^{G11}$ (plasmid) | This paper | | Generated using site-directed mutageneis with primer GGCGGGTGCTG GTGGTGAACAAGC TGGGTATGCGC |
| Recombinant DNA reagent | PGEX-Syx1A$^{\Delta C}$ (plasmid) | This paper | | Primers CGCGGATCCA TGACTAAAGA CAGATTAGCCG and TCCCCCGGG TTACATGAAATAAC TGCTAACAT were used to amplify Syx1A, which was then cloned into PGEX-4T1, site-directed mutagenesis with primer GTAAAGCCCGA CGAAAG TAGATCATGAT ACTGATC was used to remove the C-terminal tail |
| Peptide, recombinant protein | MBP-Rop$^{WT}$ /MBP-Rop$^{G11}$ | This paper | | Recombinant MBP-Rop was expressed from PMAL-Rop in Rosetta$^{TM}$ cells, purified using amylose resin, and eluted with maltose |

*Continued on next page*

*Continued*

| Reagent type (species) or resource | Designation | Source or reference | Identifiers | Additional information |
|---|---|---|---|---|
| Peptide, recombinant protein | GST-Syx1A$^{\Delta C}$ | This paper | | Recombinant GST-Syx1A$^{\Delta C}$ was expressed from PGEX-Syx1A$^{\Delta C}$ in Rosetta$^{TM}$, purified using GST resin, and eluted with glutathione |
| Commercial assay or kit | QuikChange Lightning Site-Directed Mutagenesis Kit | Agilent | 210518 | |
| Commercial assay or kit | Coomassie Blue R-250 Solution | TekNova | C1050 | |
| Chemical compound, drug | Phorbol 12-myristate 13-acetate Phorbol Ester (PdBU) | Sigma-Aldrich | Sigma-Aldrich CAS: 16561-29-8 | Stock concentration: 10 mM Final Concentration (in HL3 saline): 1 µM |
| Chemical compound, drug | Philanthotoxin-433 (PhTX) | Sigma-Aldrich (disc.) Santa Cruz Biotech. | Sigma Aldrich CAS: 276684-27-6 Santa Cruz Biotech. sc-255421 | Stock concentration: 5 mM Final Concentration (in HL3 saline): 10–20 µM |
| Software, algorithm | Sharp-electrode recordings | Molecular Devices | Clampex (10.3.1.5) | |
| Software, algorithm | EPSP analysis | Molecular Devices | Clampfit (10.3.1.5) | |
| Software, algorithm | EPSC and Pr analysis | Wave-Metrics | Igor Pro (6.3.4.1) RRID: SCR_000325 | custom script |
| Software, algorithm | RRP, train analysis | (*Müller et al., 2015*) | | |
| Software, algorithm | mEPSP analysis | Synaptosoft | Mini Analysis 6.0.7 RRID: SCR_002184 | |
| Software, algorithm | GraphPad Prism (7.0 c) | GraphPad | RRID: SCR_002798 | |
| Software, algorithm | Fiji | NIH | RRID: SCR_002285 | |
| Other | Amylose Resin | New England Biolabs | NEB: E8021 | used to purify MBP recombinant protein |
| Other | GST Bind Resin | Novagen | 70541 | used to purify GST recombinant protein and for pull-down of recombinant protein |

## Fly stocks and genetics

In all experiments, the $w^{1118}$ strain was used as the wild-type control. Animals were raised between 22–25°C. $Rop^{G27}$ (*Harrison et al., 1994*), $syx1A^{\Delta 229}$ (*Schulze et al., 1995*), $dunc$-$13^{P84200}$ (*Aravamudan et al., 1999*), and $Df(3L)BSC735$ (*Cook et al., 2012*) were obtained from the Bloomington Drosophila Stock Center. $Rop^{G11}$ (*Harrison et al., 1994*) was provided by Hugo Bellen.

$Rbp^{STOP1}$ (*Liu et al., 2011*) was provided by Stephan Sigrist. $rim^{103}$ was generated in the Davis lab as described previously (*Müller et al., 2012b*). UAS-Rop-RNAi animals were obtained from Vienna Drosophila Resource Center (VDRC) (stock GD1523). The $elav^{C155}$-Gal4 driver has been previously described (*Lin and Goodman, 1994*).

## Electrophysiology

Sharp-electrode recordings were made from muscle six in abdominal segments 2 and 3 from third-instar larvae using an Axoclamp 2B or Multiclamp 700B amplifier (Molecular Devices), as described previously (*Müller et al., 2012b*). Two-electrode voltage clamp recordings were performed with an Axoclamp 2B amplifier. Recordings were made in HL3 saline containing the following components (in mM): 70 NaCl, 5 KCl, 10 MgCl2, 10 NaHCO3, 115 sucrose, 4.2 trehalose, 5 HEPES, and 0.3 CaCl2 (unless otherwise specified). For acute pharmacological homeostatic challenge, larvae were incubated in Philanthotoxin-433 (PhTX; 10–20 uM; Sigma- Aldrich or Santa Cruz Biotechnology) for 10 min (*Frank et al., 2006*). EGTA-AM (25 uM in HL3; Invitrogen) was applied to the dissected preparation for 10 min. Recordings were excluded if the resting membrane potential (RMP) was more depolarized than −55 mV. A threshold 40% decrease in mEPSP amplitude, below average baseline, was used to confirm the activity of PhTX. After EGTA application, the preparation was washed with HL3. PdBU (1 uM) was applied to the dissected preparation and incubated for 10 min. before recording. Quantal content was estimated by calculating the ratio of EPSP amplitude/average mEPSP amplitude and then averaging recordings across all NMJs for a given genotype. EPSC data were analyzed identically. Paired pulse ratios were quantified by calculating the ratio of the 4th EPSC/1$^{st}$ EPSC. EPSP and mEPSP traces were analyzed in IGOR Pro (Wave-Metrics) and MiniAnalysis (Synaptosoft).

The RRP was estimated by cumulative EPSC analysis (*Schneggenburger et al., 1999*; *Weyhersmüller et al., 2011*; *Müller and Davis, 2012a*). Muscles were clamped at −65 mV in two-electrode voltage configuration, and EPSC amplitudes during a stimulus train (60 Hz, 30 stimuli) were calculated as the difference between peak and baseline before stimulus onset of a given EPSC. The average cumulative EPSC amplitude for a given muscle was obtained by back-extrapolating a line fit to the linear phase (the last 200 ms) of the cumulative EPSC plot to time 0. The apparent RRP size was obtained by dividing the cumulative EPSC amplitude by the mean mEPSP amplitude recorded in the same cell in current clamp mode before placing the second electrode in the muscle.

## Molecular biology

The *PGEX-syx1A$^{ΔC}$* construct was generated by PCR introduction of BAMHI/SMAI sites at the 5' and 3' ends of *syx1A* cDNA obtained from the Drosophila Genomics Resource Center (DGRC; Source cDNA: LD43943). The BAMHI-SMAI *syx1A* cDNA fragment was then cloned into the *PGEX-4T1* vector (Addgene) by standard techniques to make *PGEX-syx1A*. To remove the C-terminal membrane-bound tail of *syx1A*, the *syx1A* point mutation (A802T) was made by site-directed mutagenesis of *PGEX-syx1A*.

The *PMAL-Rop* construct was generated by PCR introduction of BAMHI/ECORI sites at the 5' and 3' ends of *Rop* cDNA obtained from the Drosophila Genomics Resource Center (DGRC; Source cDNA: SD04216). The BAMHI-ECORI *Rop* cDNA fragment was then cloned into the PMAL-c5e vector (New England Biolabs) to create *PMAL-Rop*. To generate the *PMAL-Rop$^{G11}$* construct, the *Rop* point mutation (G133A) was made by site-directed mutagenesis of *PMAL-Rop.* All site-directed mutagenesis reactions were performed using the QuikChange Lightning Site-Directed Mutagenesis Kit (Agilent).

## Recombinant protein expression and purification

Recombinant MBP-Rop, MBP-Rop$^{G11}$ (D45N) and GST-syx1A$^{ΔC}$ (amino acids 1–268) were expressed from respective expression vectors *PMAL-Rop*, *PMAL-Rop$^{G11}$*, and *PGEX-syx1A$^{ΔC}$* in Rosetta$^{TM}$ cells (Novagen). Cell pellets were lysed in column buffer (20 mM Tris (pH 7.0), 150 mM NaCl, 2 mM EDTA and 0.1% NP40) and cleared by centrifugation. MBP protein was purified using amylose resin (NEB). GST protein was purified using GST bind resin (Novagen). Proteins were eluted using maltose (for MBP proteins) or glutathione (for GST proteins) per the manufacturers recommendations.

## In vitro binding assay

GST- syx1A$^{\Delta C}$ (2 ug) proteins were bound to GST bind resin (Novagen) for 1 hr at 4°C with varying concentrations of MBP-Rop and MBP-Rop$^{G11}$ (uM): 0, 0.1, 0.2, 0.3, 0.4, 0.6, 0.8, 1, 1.5, 3, 6, and NP40 buffer (6 mM Na2HPO4, 4 mM NaH2PO4, 1% NONIDET P-40, 150 mM NaCl, 2 mM EDTA, 50 mM NaF, 4 ug/ml leupeptin, 0.1 mM Na3VO4). After washing coated beads in NP40 buffer, proteins were eluted by boiling in 5X SDS sample buffer and denatured by boiling for 10 min. Proteins were resolved by SDS-PAGE on a 4–12% Bis-Tris gel (Life Technologies), stained with Coomassie Blue (TekNova) for 30 min and de-stained with destaining solution (50% H$_2$O, 40% MeOH, 10% Acetic Acid) for 2 hr. The gel analysis tool on Image J was used to quantify the fraction of Rop protein binding to syx1A. The Kd was derived from a dose response curve that was fit to the data using Prism 6.

## Acknowledgements

We thank Dr. Ting Ting Wang for critical reading of the manuscript. Supported by NIH Grant R35NS097212 to GWD. We also acknowledge support of the Kavli Institute for Fundamental Neuroscience at UCSF.

## Additional information

### Competing interests

Graeme W Davis: Reviewing editor, *eLife*. The other authors declare that no competing interests exist.

### Funding

| Funder | Grant reference number | Author |
| --- | --- | --- |
| National Institute of Neurological Disorders and Stroke | R35NS097212 | Graeme W Davis |

The funders had no role in study design, data collection and interpretation, or the decision to submit the work for publication.

### Author contributions

Jennifer M Ortega, Conceptualization, Data curation, Formal analysis, Investigation, Writing—original draft, Writing—review and editing; Özgür Genç, Data curation, Formal analysis, Writing—review and editing; Graeme W Davis, Conceptualization, Funding acquisition, Methodology, Writing—original draft, Project administration, Writing—review and editing

### Author ORCIDs

Graeme W Davis (iD) http://orcid.org/0000-0003-1355-8401

### Decision letter and Author response

Decision letter https://doi.org/10.7554/eLife.40385.015
Author response https://doi.org/10.7554/eLife.40385.016

## Additional files

### Supplementary files

• Supplementary file 1. Summary of raw data.
DOI: https://doi.org/10.7554/eLife.40385.012

• Transparent reporting form
DOI: https://doi.org/10.7554/eLife.40385.013

### Data availability

Data generate during this study are included in the manuscript and supporting files and tables.

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
