## [Decision Letter]

Thank you for submitting your article "Molecular Mechanisms that Stabilize Short Term Synaptic Plasticity During Presynaptic Homeostatic Plasticity" for consideration by *eLife*. Your article has been reviewed by three peer reviewers, and the evaluation has been overseen by a Reviewing Editor and Eve Marder as the Senior Editor. The following individual involved in review of your submission has agreed to reveal his identity: Ruud F Toonen (Reviewer #3). Two other reviewers remain anonymous.

The reviewers have discussed the reviews with one another and the Reviewing Editor has drafted this decision to help you prepare a revised submission.

Here the authors use *Drosophila* NMJ to address the molecular mechanisms of presynaptic homeostatic plasticity (PHP), specifically motivated by understanding the basis for the observed persistence of short-term plasticity (STP) during PHP. The authors report that *Drosophila* ortholog of Unc-18, Rop, is required for PHP, and the interaction between Rim and Rop is essential. Interestingly, reducing the level of Syntaxin1A or disrupting Syntaxin1A and Rop interaction can restore PHP, which suggests that under control conditions, Syntaxin1A limits the availability of Rop needed to drive PHP. An impressive amount of data are presented, and the results are overall convincing. The main findings highlight a key interplay of Rop and Syntaxin in controlling PHP, and by extending our current understanding of the mechanisms of presynaptic plasticity, the study should be of general interest to the readership of *eLife*. However, several major concerns need to be addressed that are listed below. Aside from one new set of experiments requested (point 1 below), most of the concerns relate to premature or over-interpretation of data and inadequate reference of previous literature that should be readily addressable by careful rewriting of the text.

Essential revisions:

1) The authors discuss the release probability in the context of the currently fashionable framework of super-priming. However, using only the effect size of PdBu on EPSC amplitude is at best an indirect method to access the release probability of two postulated pools of release ready vesicles, whose existence have not been carefully investigated at the *Drosophila* NMJ. Note, for example that in contrast to Taschenberger et al., 2016, the fourth EPSC is still strongly increased with PdBu (Figure 1A and 7A). One should compare the time course of synaptic depression in the presence of PdBu using long stimulus train (cf. Figure 5D).

2) Although the manuscript contributes new proteins important for PHP and places Rop in a primary position, the precise molecular mechanism by which Rop fulfills such a role is not entirely clear. The authors argue that due to lack of regulation of RRP size, as Unc18-1 does in mammalian systems, Rop is "essential for the expansion of the super-primed vesicle pool during expression of PHP". They show strong potentiation of PdBu application in the Rop/Rim heterozygote, consistent with such a role, but in this experiment it is not clear whether the effect is contributed by Rop or Rim. To support their claim of Unc18-1 being a regulator of the super-primed vesicle pool, the authors should show PdBu effect on single Rop heterozygotes and/or address this issue better in the Discussion. The rescued miniature EPSP frequency (Figure 8F) could play a role in rescuing PHP, and it is unclear if evoked release is also rescued.

3) The conclusion that PHP is mainly mediated by an increase in the RRP was first shown in Weyhersmüller et al., 2011. This should be cited at the appropriate locations in the text (e.g. Introduction, third paragraph, subsection “Rop dependent vesicle priming correlates with expression of PHP”, second paragraph,, subsection “PHP potentiates release without altering release dynamics”, first paragraph), "But, current models have yet to address how short-term release dynamics are stabilized" (cf. Figure 8 of Weyhersmüller et al., 2011). Moreover, although not explicit, the concept similar to super-priming has been discussed by several papers before Taschenberger et al., 2016 such as Mueller et al., 2010.

4) The specificity of the rescue of Rop PHP phenotype by Syntaxin1A requires clarifications. That compound heterozygotes of Rop and Syntaxin rescue the PHP defect of Rop single heterozygotes is an unexpected and interesting finding. However, the authors also report several other compound heterozygotes (Rop/Munc13, Rop/BRP) that rescue the PHP defect of Rop single heterozygotes arguing against a specific role for syntaxin interaction in controlling Rop-dependent PHP. Could the authors comment on these findings in the Discussion?

5) In the Rop mutants, the mini frequency (Figure 3) and the amplitude of evoked EPSCs (Figure 4) were altered. The authors argue for a "specific activity of Rop necessary for PHP", but it is not surprising that interference with Unc18 (one of the most needed proteins to build functional release sites) impairs spontaneous and evoked release as well as PHP.

---

## [Author Response]

Essential revisions:1) The authors discuss the release probability in the context of the currently fashionable framework of super-priming. However, using only the effect size of PdBu on EPSC amplitude is at best an indirect method to access the release probability of two postulated pools of release ready vesicles, whose existence have not been carefully investigated at the Drosophila NMJ. Note, for example that in contrast to Taschenberger et al., 2016, the fourth EPSC is still strongly increased with PdBu (Figure 1A and 7A). One should compare the time course of synaptic depression in the presence of PdBu using long stimulus train (cf. Figure 5D).

We thank the reviewers for this comment and suggestion. A) We have completed an analysis of synaptic transmission using longer stimulus trains in the presence and absence of PdBu and have added these data (new Figure 1—figure supplement 1). These data nicely agree with the results of Taschenberger et al., 2016, showing that release converges to a similar steady-state release rate in the presence and absence of PdBu during a stimulus train. These data also support the conclusion that PdBu is primarily affecting vesicle release during the early phase of the stimulus train. We have added this information to the Results section of our paper, again citing the work of Taschenberger et al., 2016; see Results, fifth paragraph.

2) Although the manuscript contributes new proteins important for PHP and places Rop in a primary position, the precise molecular mechanism by which Rop fulfills such a role is not entirely clear. The authors argue that due to lack of regulation of RRP size, as Unc18-1 does in mammalian systems, Rop is "essential for the expansion of the super-primed vesicle pool during expression of PHP". They show strong potentiation of PdBu application in the Rop/Rim heterozygote, consistent with such a role, but in this experiment it is not clear whether the effect is contributed by Rop or Rim. To support their claim of Unc18-1 being a regulator of the super-primed vesicle pool, the authors should show PdBu effect on single Rop heterozygotes and/or address this issue better in the Discussion. The rescued miniature EPSP frequency (Figure 8F) could play a role in rescuing PHP, and it is unclear if evoked release is also rescued.

We thank the reviewers for these comments. Several points have been made here and we have addressed all of them through the addition of new data to our manuscript.

A) First, we have added new information to the manuscript regarding PdBu application to the individual heterozygous mutants *Unc18/+* and *Rim/+*. We show that Unc18/+ and Rim/+ each show wild type potentiation of release in the presence of PdBu (new Figure 7C and D; Results subsection “Loss of Rop and Rim limits the super-primed vesicle pool”). This is consistent with each individual heterozygous mutant having a mild effect on baseline neurotransmission alone (see Figure 6A-C). But, when the two mutations are combined as a double heterozygous animal (*Rop/Rim*), there is a strong synergistic reduction in baseline transmission (see Figure 7A), a complete block of PHP (see Figure 6), and a synergistic increase in the effect of PdBu application (see Figure 7A-C). This type of synergistic genetic interaction is generally taken as evidence that two genes participate in the same process. It is not possible to make any conclusion regarding whether these genes are acting in a linear signaling pathway or in parallel signaling pathways. But, based on formal genetic argument, we are able to conclude that they likely converge to control the same processes at the presynaptic terminal, both the PdBu sensitive vesicle pool and PHP. This is the same argument that we made in our initial submission, but it is now supported by a more extensive data set and we thank the reviewers for prompting us to do this additional control. We have updated our Results section to include these data (new Figure 7C, D). We have added the following text to the manuscript, “This type of synergistic genetic interaction is generally taken as evidence that two genes participate in the same process. It is not possible to make any conclusion regarding whether these genes are acting in a linear signaling pathway or in parallel signaling pathways. But, based on formal genetic argument, we are able to conclude that they likely converge to control the same processes at the presynaptic terminal, both the PdBu sensitive vesicle pool and PHP.”

B) The reviewer is correct that there is a correlation between the rescue of mini frequency and the rescue of PHP. We now make note of this fact in our Results section (subsection “Loss of *syx1A* rescues PHP in the heterozygous *rop* mutant background”, last paragraph). There is, as yet, no indication that mEPSP frequency is directly responsible for the induction of PHP (Frank et al., 2006, 2009; Harris et al., 2015; Goold and Davis, 2007). Rather, PHP is induced by a change in mini amplitude. Thus, we favor the conclusion that the rescue of mini rate reflects the restoration of the docked/primed vesicle pool and that this restoration supports the full expression of PHP.

C) The reviewer states that it is, “unclear if evoked release is also rescued in Figure 8”. In our original submission, we quantify mEPSP amplitude, EPSP amplitude and quantal content for each of the genotypes in Figure 8 (Figure 8B, C and D respectively). The reviewer must simply have missed these data during the review process.

3) The conclusion that PHP is mainly mediated by an increase in the RRP was first shown in Weyhersmüller et al., 2011. This should be cited at the appropriate locations in the text (e.g. Introduction, third paragraph, subsection “Rop dependent vesicle priming correlates with expression of PHP”, second paragraph, subsection “PHP potentiates release without altering release dynamics”, first paragraph), "But, current models have yet to address how short-term release dynamics are stabilized" (cf. Figure 8 of Weyhersmüller et al., 2011). Moreover, although not explicit, the concept similar to super-priming has been discussed by several papers before Taschenberger et al., 2016 such as Mueller et al., 2010.

We thank the reviewers for pointing us to Figure 8 in Weyhersmuller et al., 2011. Indeed, they show that paired-pulse stimulation is unaltered following the induction of PHP in their study. We have added this reference to the second paragraph of our Introduction, where we first make the statement that presynaptic release dynamics remain stable during PHP. It appears that this is the first clear demonstration of this phenomenon and we are happy to recognize this in our Introduction. We also reference Mueller et al., 2010, when we refer to super-priming in the Introduction. Thank you for pointing this out.

4) The specificity of the rescue of Rop PHP phenotype by Syntaxin1A requires clarifications. That compound heterozygotes of Rop and Syntaxin rescue the PHP defect of Rop single heterozygotes is an unexpected and interesting finding. However, the authors also report several other compound heterozygotes (Rop/Munc13, Rop/BRP) that rescue the PHP defect of Rop single heterozygotes arguing against a specific role for syntaxin interaction in controlling Rop-dependent PHP. Could the authors comment on these findings in the Discussion?

There seems to be some confusion on the part of the reviewers. We never analyzed the *Rop/Munc13* genetic combination, and we never analyzed *Rop/Brp* genetic combination (in fact we never analyzed any genetic combination with Brp in the current study). We think that the reviewers are referring to data presented in Figure 6. Several points should be emphasized:

A) In Figure 6 we show a strong, synthetic genetic interaction of *Rop/+* with *Rim/+*. This synthetic genetic interaction is revealed as a large decrease in baseline release and a complete block of PHP. This is a very strong genetic interaction and we performed several experiments to underscore just how unique this genetic interaction really is. Therefore, in panels F, G and H, we tested known binding partners of RIM in order to probe the specificity of the synthetic genetic interaction of *Rop/+* with *Rim/+*. These experiments *do not* relate to the genetic interaction of *Rop/+* with *Syntaxin/+*. They relate to the genetic interaction of Rop and RIM.

B) With respect to data presented in panel F. Rim Binding Protein is known to bind RIM. Because *Rop/+* interacts strongly with Rim/+, we asked if *Rop/+* also shows a strong interaction with *RBP/+.* It does not. Homeostatic plasticity is robustly expressed. Thus, there is something unique about the genetic interaction of *Rop/+* with *Rim/+*.

The reviewers may have noticed that at PHP is no different when the *Rop/RBP* double heterozygous animal is compared to either Rop/+ alone or when compared to wild type. This ambiguity is due to increased variability in the *Rop/RBP* double heterozygous mutant. Thus, no conclusion can be made regarding rescue in the double heterozygous condition. But, we are able to conclude that the *Rop/RBP* does not show the same type of strong synthetic block of PHP as observed in the *Rop/rim* double heterozygous animal. For purposes of clarity, we have added the following sentence to the Results section, “It is worth noting that PHP is highly variable in the *Rop^G27^/ rbp^STOP1^* double heterozygous mutant, preventing us from making any additional conclusions regarding whether PHP is similar to, or different from wild type.”

C) In panel F, we test the genetic interaction of *Rim/+* with *Unc13/+*. Again, Unc13 is a known binding partner of RIM. Again, PHP is robustly expressed, underscoring the uniqueness of the genetic interaction of Rop and RIM.

D) In panel G, we test the genetic interaction of *Rim/+* with *RBP/+.* Again, PHP is robustly expressed, underscoring the uniqueness of the genetic interaction of Rop and RIM.

E) It is somewhat surprising that the *rim/+; unc13/+* double heterozygous mutant has significantly more PHP than observed in *rim/+* alone. This could be informative regarding how RIM functions at the synapse, but this does not relate directly to the function of Rop, or the rescue of Rop by loss of Syntaxin. It could be something to follow up in future experiments, but it would be an entirely new line of investigation aimed at understanding the interaction of RIM with Unc13 during PHP. We have added the following sentence to our Results, “It is somewhat surprising that the *rim/+; unc13*/+ double heterozygous mutant has significantly more PHP than observed in *rim*/+ alone, perhaps relating to the functional importance of the RIM-Unc13 biochemical interaction during PHP. Regardless, when taken together, our results underscore the specificity and importance of the genetic interaction between *rop* and *rim* and the relevance of rop to the mechanisms of PHP.”

We have re-read the text in this portion of the manuscript and believe that we are clear about the purpose and interpretation of our results. We think that this may simply have been a misunderstanding on the part of the reviewers. We acknowledge that there are many genes and genetic interactions presented and this can be confusing if a reader is moving rapidly between figures. But, this is a genetic analysis and we see no way around this issue since it is important to label all genotypes precisely on the individual graphs.

5) In the Rop mutants, the mini frequency (Figure 3) and the amplitude of evoked EPSCs (Figure 4) were altered. The authors argue for a "specific activity of Rop necessary for PHP", but it is not surprising that interference with Unc18 (one of the most needed proteins to build functional release sites) impairs spontaneous and evoked release as well as PHP.

We do not state that Unc18 is specific for PHP. We state that Unc18 has *an activity* that is specific for PHP. This sentence refers to one activity of Unc18, among many. Indeed, the full text of our argument in this portion of the discussion should make this quite clear, since the very next sentence discusses the role of Unc18 during baseline release. “We provide several lines of evidence supporting the conclusion that Unc-18 has a specific activity necessary for PHP. For example, neurotransmitter release in the heterozygous *Rop^G27^* /+ mutant remains highly sensitive to changes in extracellular calcium. Yet, across a 10-fold range of extracellular calcium, PHP is suppressed by a constant fraction of ~30%. We also pursued a series…” Thus, we believe we were quite clear. But, after re-reading this section, we see no need for the use of the word ‘specific’ in this context and have deleted it from that sentence.